*The Company of*
**Biologists**

## RESEARCH ARTICLE

# Sunlight surveillance: a simplified approach for the monitoring of harmful ultraviolet radiation in freshwater ecosystems

Coen Hird*, Rebecca L. Cramp and Craig E. Franklin

## ABSTRACT

Ultraviolet radiation (UVR) has a range of strong effects on freshwater ecosystems, and changing UVR is implicated in global amphibian declines. The link between UVR and amphibian declines is not well understood, largely due to limited understanding of actual UVR exposure regimes in freshwater ecosystems. Logistical challenges in measuring aquatic UVR regimes *in situ* have impeded progress, including the design of cost-effective radiometric monitoring tools and the measurement of UVR at ecologically relevant scales. We designed novel underwater UVR loggers and deployed them in southeast Queensland creeks to log near-continuous underwater ultraviolet index (UVI) for 11 days across four sampling events within the peak amphibian breeding season. We compared these data with solar modelling, dosimetric and handheld radiometric techniques. The dataset (2616 h from 39 UV sensors) revealed a highly heterogenous daily UVR microenvironment that showed capacity for harmful UVI exposures at both sites. Traditional cumulative or periodic UVR monitoring approaches often missed these acute high UVR exposures. Fine scale UVR data logging was proven to be a cost-effective approach for monitoring the UVR microenvironment in freshwater systems. While no single method can be considered universally optimal, our results highlight the advantages of continuous *in situ* UV monitoring, particularly for capturing short-duration ultraviolet fluctuations relevant to aquatic organisms.

KEY WORDS: Ultraviolet index (UVI), Temperature, Ozone hole, Climate change, Ultraviolet, Data loggers

## INTRODUCTION

Ultraviolet radiation (UVR) is a key environmental variable shaping the evolution of life, largely due to the detrimental effects of high UVR on biological structures like DNA (Cockell, 2000; Cockell and Blaustein, 2001). Although UVR exposure has wide-ranging effects on organisms and biological processes, not all wavelengths of UV reach the Earth's surface. The sun emits UV in three major wavelength groups: UVA (315–400 nm), UVB (280–315 nm) and UVC (100–280 nm). Stratospheric ozone ($O_3$) absorbs short-wave UVC and most of UVB, but not UVA. Although the majority of UVB is absorbed by the stratospheric ozone column, biologically

School of the Environment, The University of Queensland, Brisbane 4072, Australia.

*Author for correspondence (c.hird@uq.edu.au)

C.H., 0000-0001-9812-4818; R.L.C., 0000-0001-9798-2271; C.E.F., 0000-0003-1315-3797

significant levels of solar UVB reach the Earth's surface. Incident UVR varies spatiotemporally (Bais et al., 2015), driven by multiple interacting factors such as ozone column length, solar angle, cloud and canopy cover, and surface reflectance (albedo) (Koepke et al., 2002; Belmont et al., 2009). Elevation and latitude are important variables influencing incident UVR levels in aquatic environments (Körner, 2003; Diamond et al., 2005; Wang et al., 2014). UVB can increase by 8–10% per kilometre above sea level (Madronich et al., 1995; Pfeifer et al., 2006; Dahlback et al., 2007), and is lowest at the poles and highest at the equator (Caldwell et al., 1980; Barnes et al., 1987). UVR can penetrate significantly into aquatic systems, particularly those with clear water and little overhanging vegetation (Xenopoulos and Schindler, 2001). In clear waters, UVB has been recorded penetrating to depths of 10–20 m, while more turbid or high dissolved organic matter (DOM) environments, attenuation can occur within the first few centimeters to meters (Scully and Lean, 1994; Häder et al., 2015). For instance, in some freshwater lakes, UVA can reach depths of 5–30 m, whereas UVB penetration is typically limited to 1–5 m depending on DOM concentrations (Morris et al., 1995). DOM concentrations can vary substantially across waterbodies (Schindler et al., 1992; Morris et al., 1995; Williamson et al., 1996; Gergel et al., 1999), resulting in different UVR penetrance across environments. Aquatic habitats in montane ecosystems are typically associated with higher UVR penetrance due to lower aquatic DOM concentrations (Häder et al., 2007).

In addition to abiotic drivers of global UVR patterns, anthropogenic factors have influenced levels of UVR reaching the Earth. As a result of stratospheric ozone depletion by the emission of industrial chemicals such as chlorofluorocarbons, UVB levels have increased by 2–6% in some areas since last century (Lemus-Deschamps and Makin, 2012) and will likely remain high throughout the 21st century (Montzka et al., 2018). Though the effects of ozone depletion on UVR reaching the Earth's surface have been largely mitigated by the Montreal Protocol (World Meteorological Organization, 2022), UVR is predicted to change significantly in freshwater ecosystems because of climate change, as changes such as cloud cover, extreme weather events and vegetation responses modify incident UVR (Williamson et al., 2014; Bais et al., 2018; Barnes et al., 2019; McKenzie et al., 2020). This is important because UVR is a critical determinant of health in freshwater organisms (Peng et al., 2017), and has been implicated in the global amphibian extinction crisis (Blaustein et al., 1995; Blaustein et al., 2003; Häder et al., 2015).

Globally, amphibians face one of the highest extinction risks of any vertebrate clade (Houlahan et al., 2000; Hof et al., 2011). Over the past few decades, amphibian extinction rates exceeded and will continue to exceed background rates by over four orders of magnitude (McCallum, 2007; Alroy, 2015). Many of these declines were classed as enigmatic (Stuart et al., 2004), but now most of these have been largely linked to the emergence and spread of the pathogenic amphibian chytrid fungus *Batrachochytrium*

*dendrobatidis* (Bd) (Berger et al., 1998; Young et al., 2001; Gillespie et al., 2020). The links between environmental drivers of amphibian declines and disease is an active area of research, and multiple abiotic factors have been implicated with amphibian disease occurrence and severity including temperature, rainfall, and changing levels of UVR (Carey, 1993; Broomhall et al., 2000; Kiesecker et al., 2001). In Australia, amphibian declines have primarily occurred across a significant latitudinal gradient along the eastern coastline, with a disproportionately high number of declines found at higher elevations >400 m above sea level associated with cooler temperatures and higher levels of UVR (Biodiversity Group, 1999; Blaustein and Wake, 1990; Bradford, 1991; Kiesecker et al., 2001; McDonald and Alford, 1999; Richards et al., 1993; Young et al., 2001).

Increased UVR is hypothesised to influence amphibian populations by negatively affecting eggs and tadpoles. Amphibian eggs are often laid in shallow water with high UV exposure, and tadpoles are typically diurnal, active during spring and summer when UVR levels are highest. Additionally, both life stages have limited ability to avoid UVR, making them particularly vulnerable (Blaustein et al., 2003; Lundsgaard et al., 2023). Environmental temperature is a significant determinant of UVR-associated impacts in amphibian larvae. Temperature has profound effects on biochemical reaction rates and physiological function, particularly for ectotherms where physiological rates are directly related to organismal performance (Huey and Kingsolver, 1989; Angilletta, 2009). The negative effects of UVR on amphibian health are compounded when UVR exposures occurs at low temperature (Kiesecker and Blaustein, 1995; van Uitregt et al., 2007; Bancroft et al., 2008a; Alton and Franklin, 2012; Morison et al., 2020; Lundsgaard et al., 2020; Hird et al., 2022). These negative effects were hypothesised to be caused by the depressive effects of temperature on photoenzymatic DNA repair rates (Morison et al., 2020; Hird et al., 2022). While studies using sensors to characterise thermal microenvironments experienced by organisms are numerous (Humphreys, 1978; Stelzner and Hausfater, 1986; Jimenez et al., 2008; Pincebourde et al., 2016), fewer studies have attempted to build a comparable picture of the UVR microenvironment in freshwater ecosystems (Sommaruga and Psenner, 1997; Bukaveckas and Robbins-Forbes, 2000; Markager and Vincent, 2000; Laurion et al., 2000). An understanding of the UVR microenvironment in freshwater ecosystems and the UVR doses that amphibians experience *in situ* is lacking (Licht, 2003).

The spatiotemporal correlation between high elevation amphibian declines and anthropogenic increases in UVR is likely driven in part by disease-related thermal dynamics (e.g. pathogens preferring cooler temperatures such as those at higher elevation). However, a mechanistic basis for the spatiotemporal correlation between high elevation amphibian declines and anthropogenic increases in UVR remains unclear. Data from mesocosm and laboratory-based studies have shown that the larvae and embryos of many ectotherms are highly sensitive to UVR (Alton and Franklin, 2017; Peng et al., 2017; Downie et al., 2023), yet the hypothesis that lab-based effects are replicable in the field environment remains controversial, largely because measurement of ecologically and physiologically relevant UVR levels in amphibian habitats has been difficult and rarely attempted. Although many juvenile and adult amphibian life stages are nocturnal, most aquatic larval stages are diurnal and may actively seek out sunlight to thermoregulate (selecting preferred water temperatures), potentially exposing them to significant UVR doses (Brattstrom, 1979; Bradford, 1984; Wollmuth and Crawshaw, 1988; Ultsch et al., 1999; Bancroft et al.,

2008b). Likewise, embryos may receive significant UVR doses if oviposition sites receive significant sun exposure. Understanding ecologically relevant UVR doses in amphibian habitats is challenging due to the highly variable nature of UVR attenuation. Satellite estimation of UVR only gives broad scale UVR measurements and is prone to error (Bais et al., 2015). Because individual animals experience climate at fine spatial scales, climate heterogeneity ultimately determines the microclimates that organisms will experience (Pincebourde et al., 2016). For this reason, understanding the UVR that amphibians may receive in nature requires fine scale temporal and spatial measurements of UVR.

Considering that UVR is a critical determinant of health in freshwater organisms, understanding how UVR will change and interact with other environmental variables is critical to predict how climate change will impact freshwater ecosystems. There is a critical need to develop novel and feasible ways to quantify ecologically realistic UVR exposure regimes in freshwater ecosystems to accurately monitor changing UVR in the future. In this study, we used bespoke low cost UVR and temperature loggers, handheld spectroradiometry, and UVR dosimeters to generate a unique dataset that characterises the UV environment in two amphibian habitats in southeast Queensland (Australia) over the amphibian breeding period. We hypothesised that traditional monitoring techniques would provide coarse estimates of natural UVR exposure regimes compared with fine-scale UVR measurements obtained with UVR data loggers. Furthermore, we expected traditional techniques would miss high UVR exposures potentially harmful to aquatic life. These data will be important for understanding the relevance of past and future laboratory-based studies investigating how freshwater organisms respond to UVR, and ultimately predicting how aquatic ecosystems will respond to global climate change.

## RESULTS
Biologically significant levels of UVR reached underwater sensors as well as dosimetric film at Guanaba Creek (Guanaba Indigenous Protected Area), which had clear waters across the study period. This was also true at Boy-Ull Creek (Springbrook National Park), though UVR levels were generally lower, water temperatures cooler, and the number of amphibian species recorded during surveys was higher at Boy-Ull Creek compared with Guanaba Creek.

### Water UV transmittance
Percent water transmittance was significantly higher in the UVA band compared to UVB bands ($F_{2,42}$=3.34, $P$=0.045), but there was no statistically significant difference between sites ($F_{3,42}$=1.45, $P$=0.241; Table 2). Percent UV transmittance averaged over 97% at all sites and all wavelengths. The high transparency was attributed to the clarity of the water (Laurion et al., 2000). The maximum pool depth at both sites occurred in the centre of the stream where flows were highest and did not exceed 40 cm depth. In the still areas of the stream where the local amphibian species are more likely to oviposit, average depths were <20 cm. At 20 cm below the water surface, Beer–Lambert's equation estimated UV irradiances 78–96% of water surface UVA and 55–65% of water surface UVB. While UV levels at 1 cm under the surface were therefore higher than animals might experience at the substrate, eggs and early larvae of local species are often laid or found at the water surface. Furthermore, tadpoles typically exploit the full water column, such as individuals frequently surfacing to breathe. From this, we assumed that UVA measurements at the water surface approximated

conditions throughout the water column, whereas UVB attenuation was significant but still penetrated the water at depth. Any assumption that surface measurements approximate conditions at the benthos should therefore be interpreted cautiously, particularly at the most energetic UVB wavelengths, where some variability in transmittance was recorded (Table 2).

### UVR and temperature data loggers

UVI data followed daily patterns typical of global solar radiation, with higher values around solar noon as predicted in the solar models. However, considerable variation was demonstrated in the amount of UV reaching the surface of the water (Fig. 4).

Average hourly UVI was typically higher at Guanaba Creek (Guanaba Indigenous Protected Area) compared to Boy-Ull Creek (Springbrook National Park). Both sites consistently averaged levels significantly lower than solar radiative transfer model predictions under full sun no canopy cover conditions. However, maximum UVI values were often close to high UVI levels predicted under the solar models (Fig. 5). Daily maximum UVI readings were higher at Guanaba Creek than Boy-Ull Creek ($F_{1,16}$=6.24, $P$=0.024; Fig. 6). Average water temperature at Boy-Ull Creek was significantly lower than at Guanaba Creek across all months, but the magnitude of the difference depended on the month ($F_{3,52}$=4.87, $P$<0.01; Table 3).

### Spectroradiometry

There was a discrepancy between measured irradiance values on the two UVB meters, typical of non-standardised instruments calibrated at specific nominal wavelengths (Larason and Cromer, 2001; Gehrmann et al., 2004; Eppeldauer, 2012). However, Solarmeter 6.0 UVB readings were highly correlated with ILT1400 radiometer (IL1400BL, International Light, MA, USA; $F_{1,8}$=943.7, $P$<0.001), $R^2$=0.99; Fig. 7B), as well as with Solarmeter 6.5 UVI readings ($F_{1,33}$=514.4, $P$<0.001, $R^2$=0.94; Fig. 7A).

### Dosimetry and curve analyses

UVI daily dose calculated as area under the curve (AUC) were significantly higher at Guanaba Creek sites than Boy-Ull Creek sites, but the effect depended on the sampling month ($F_{4,99}$=7.12, $P$<0.001; Fig. 8). UVI daily dose calculated as change in absorbance in dosimeter film following UV exposure were also significantly higher at Guanaba Creek than Boy-Ull Creek ($F_{3,30}$=18.89, $P$<0.001; Fig. 9A), and doses were greatest in November compared to other sampled months ($F_{3,30}$=6.56, $P$<0.01; Fig. 9B).

### Amphibian surveys

Seven amphibian species were detected across the sampling periods at sites in Guanaba Creek and Boy-ull Creek where UV data loggers were deployed (Table 4), including one species listed as Endangered under the IUCN Red List (*Mixophyes fleayi*) and one species listed as Endangered under Queensland's Nature Conservation Act 1992 (*Litoria pearsoniana*). All larval amphibians recorded in visual detection surveys were found at depths from 1 cm to approximately 20 cm deep. Tadpoles were observed surfacing.

### DISCUSSION

Considerable variation was recorded in the amount of UVR reaching the water surface at Guanaba Creek (Guanaba Indigenous Protected Area) and Boy-Ull Creek (Springbrook National Park), typical of within- and between- day UVR variability recorded in previous studies measuring surface UVR (Di Sarra et al., 2002; Alados-Arboledas et al., 2003). To our knowledge, these data represented the first investigation of the UVR microenvironment in Australian freshwater ecosystems using fine-scale near-continuous radiometric monitoring. These data provided a baseline for future monitoring during the amphibian breeding season, which is useful for the detection of climate change impacts on UVR exposure regimes in aquatic systems. There has been some attention given to the capacity for potentially harmful UVR exposures across amphibian oviposition sites in the American northwest (Palen et al., 2002; Peterson et al., 2002; Diamond et al., 2005; Hossack et al., 2006). Observations at these sites captured little daily temporal variation in UVR by taking single sensor measurements and estimating UVR irradiance at single points in time, or daily doses. Many amphibian UVR field experiments were not concerned with estimating the duration of exposure to UVR in addition to irradiance (Blaustein and Wake, 1995; Hays et al., 1996; Rader and Belish, 1997; Leech and Williamson, 2001; Palen et al., 2002; Anzalone et al., 2008), which is particularly relevant in ecosystems with larger degrees of variability in UVR reaching the water surface. Considering UVR intensity can be a more important predictor of detrimental health effects in amphibians than dose alone (Lundsgaard et al., 2021), studies that estimate single mean daily UVR doses in water bodies could miss important capacities for harmful UVR exposures.

Fine-scale spatial variation in UV exposure is an important but often overlooked factor in aquatic ecosystems. While our study focused on UV measurements at a single depth (1 cm), we recognise that UVR exposure likely varies significantly within each pool due to differences in canopy cover and water depth. For example, shallow margins, open channels, and sunlit patches within the same pool may receive considerably higher UV doses than shaded microhabitats beneath overhanging vegetation or within deeper sections of the water column. We observed larvae of multiple species during surveys in pockets of full sunlight for the full duration of the sampling period (15 min). Furthermore, we recorded surface-spawning species (Limnodynastids) present at the site, but did not observe spawn during the sampling periods. This represented a capacity for potentially harmful UVR exposures in amphibian larvae at these sites throughout the breeding season, in the absence of behavioural avoidance of UVR. It is well established that high-intensity UVR exposure can cause a range of physiological impacts in amphibian larvae, including DNA damage, immunosuppression, and oxidative stress (Blaustein et al., 2003; Morison et al., 2020). However, the exact UVR sensitivity of the larval amphibians detected at these sites has not been investigated. Future studies should integrate spatially explicit UV mapping with direct observations of amphibian microhabitat use to determine whether larvae preferentially occupy low-UV refugia or are inadvertently exposed to harmful UV doses.

Although deeper logger placements could provide more direct measurements of benthic UV exposure, the Beer–Lambert law estimates of UVR attenuation with depth indicate that UVA irradiance remains highly consistent (78–96% of surface levels) throughout the shallow water column, and although UVB attenuation is likely, it does not eliminate exposure risk at the depths available in these study sites. Given that amphibian oviposition sites were primarily <20 cm deep, our measurements at 1 cm depth capture ecologically relevant UVR conditions. Furthermore, placing loggers at 1 cm ensured that we did not underestimate potential UV risk, particularly in the UVA range. Future studies could expand on these findings by deploying sensors at multiple depths to further refine depth-dependent UV exposure profiles in similar clear-water environments. We also acknowledge that some degree of change in optical properties of water samples over time was possible but given the low DOM content of these clear-water systems, we expect only minimal alteration in UV transmittance. We also note that similar post-collection storage

methods have been used in previous UVR studies in aquatic environments (Laurion et al., 2000; Palen et al., 2002; Belmont et al., 2009; Aukes et al., 2021).

In the current study, UVI values of 7 (high erythemal risk; Vanicek et al., 2000) corresponded with spectroradiometer (IL1400BL) readings around 70 µW cm$^{-2}$ UVB. These levels are within the range of UVB irradiances used in our laboratory studies that demonstrate UV sensitivity in lowland amphibian species (Kern et al., 2014; Ceccato et al., 2016; Morison et al., 2020; Lundsgaard et al., 2020, 2021, 2022; Cramp et al., 2022; Hird et al., 2022, 2023a). Laboratory studies have also highlighted the thermal sensitivity of effects of UVR on amphibian health (van Uitregt et al., 2007; Morison et al., 2020; Hird et al., 2022). Our data provided evidence that the thermal regimes used in lab experiments testing the thermal sensitivity of UVR responses on amphibian species (i.e. from 10–30°C) are within an ecologically relevant range of freshwater stream temperatures at our study sites. UVI values greater than 7 occurred across 100% of the total days at Guanaba Creek and across 55% of days at Boy-Ull Creek, probably driven by differences in canopy and cloud cover. Weather data were not directly recorded at our study sites; however, we obtained local weather data, including daily global solar exposure and cloud cover, from the Canungra Defence weather station. While this dataset provided some insight into broader atmospheric conditions, it did not resolve fine-scale differences between our two creek sites. This limitation underscores the importance of site-specific cloud cover monitoring in future studies to better account for localised variability in UVR exposure.

Importantly, average hourly UVI levels never exceeded 7 across the study period, meaning that the capacity for harmful UVR exposure was hidden when failing to account for fine scale UVI variation in these systems. These results showed that a consideration of temporal variability in UVR exposure, particularly maximum UVR exposures over fine temporal scales, is critical to understanding if freshwater systems are receiving biologically significant UVR exposures. The consequence of amphibian larval exposure to high irradiances of UVR for short periods <30 min is not yet understood. Future studies addressing the mechanistic UVR impacts on freshwater systems could replicate the conditions of our study sites by incorporating lower average doses of UVR which feature short, high intensity bursts of UVR throughout the exposure period. This would generate more ecologically realistic UVR exposures for similar ecosystems. A wider understanding of daily UVR variation in other aquatic systems would allow for the design of laboratory UVR exposure regimes that matched the organism's environment, and potential for UVR exposure. The finding that Guanaba Creek had a higher capacity for UVR exposure than Boy-Ull Creek made sense considering differences in canopy cover between sites but not when considering differences in elevation. If UVR is implicated in high elevation declines, cooler water temperatures may be a more important determinant of UVR effects on amphibian health than UV levels if higher UVR at greater elevations does not translate to higher UVR in the microenvironment. Alternatively, species/populations in high elevation/cool environments may have differential physiological or behavioural phenotypes that influence their susceptibility to UVR.

The maximum measured UVI data we collected largely agreed with the limits predicted by radiative transfer models. Considering the accuracy of clear day radiative transfer models (Antón et al., 2009), this validated our field logger measurements. As expected, clear day radiative transfer models significantly overestimate UV exposure for environments with canopy cover, so incorporation of other environmental variables into models is a promising way to estimate more realistic UV exposure regimes in aquatic systems (Diamond

et al., 2005). For example, attenuation coefficients for sites must be calculated and modelled from DOC in darker waters, though high UV transmittance of the waters obviated this requirement in the current study. Future comparative work is needed to ground truth more elaborate environmental radiative transfer models with fine-scale UV monitoring, though arguably the deployment of cost-effective UV data loggers could present an easier option than modelling.

Future climate projections suggest that changes in cloud cover, extreme weather events, and vegetation dynamics will significantly alter UV exposure in freshwater habitats. Climate models indicate that many regions, including parts of Australia, are expected to experience more frequent and prolonged periods of intense sunlight due to shifts in atmospheric circulation and reduced cloud cover (Williamson et al., 2014; McKenzie et al., 2020). Additionally, vegetation dieback due to drought and increased wildfire frequency may lead to greater canopy openness, further amplifying UV exposure in aquatic ecosystems (Barnes et al., 2019). These environmental changes could exacerbate UV stress on amphibian populations, particularly for species reliant on shaded microhabitats or those already facing other climate-related stressors such as temperature fluctuations and disease susceptibility. Understanding the fine-scale UVR dynamics in freshwater systems, as explored in this study, is therefore critical for predicting how amphibian populations may respond to future environmental change.

Obtaining UVR data in freshwater systems is inherently challenging due to the lack of accurate UV data loggers equipped for monitoring UVR in aquatic environments (Lemus-Deschamps et al., 1999; Diamond et al., 2005). Our study compared fine scale UVR logging to other coarser measurements of UVR. The finding that fine-scale monitoring of UVR detects a capacity for harmful UVR exposure missed by traditional UVR monitoring methods in freshwater systems suggested that the implementation of similar UVR data loggers in freshwater systems is critical for future monitoring of UVR in a changing climate. While handheld radiometric UVR measurements accurately captured point-in-time UVR exposures, it was not logistically possible to monitor UVR across the field sites simultaneously at the temporal scale of the data loggers (15 s intervals). Daily UVI doses estimated from logger data typically agreed with doses estimated from the dosimetry film, suggesting higher UVR doses occurred at Guanaba Creek compared to Boy-Ull Creek. However, the magnitude of the effect detected by the film was weaker and provided little insight into the potential for harmful UVR irradiances. We argue that for these reasons, studies seeking to understand the potential for harmful UVR exposures should strive to understand the variation in maximum UVI exposures in aquatic systems, ideally through logging UVR irradiance at fine temporal scales. We demonstrated that cost effective UVR loggers are a potential solution.

A significant update to our design might be the inclusion of anti-fouling coatings or external motorised wipers to prevent biofouling (e.g. Zhu et al., 2021), allowing extended field use with less frequent upkeep, enabling deployment at larger spatial scales. Furthermore, re-engineering of the electrical circuit would be recommended to make the system smaller and to simplify and extend the low power design. We recommend investment in adequate waterproofing to ensure the longevity of the instrument, such as using military grade materials, though good waterproofing designs can come with a cost trade-off.

## MATERIALS AND METHODS
### Study sites
UV Index and temperature data were collected at two sites (pools) within two separate watersheds in southeast Queensland, Australia (Fig. 1) over the

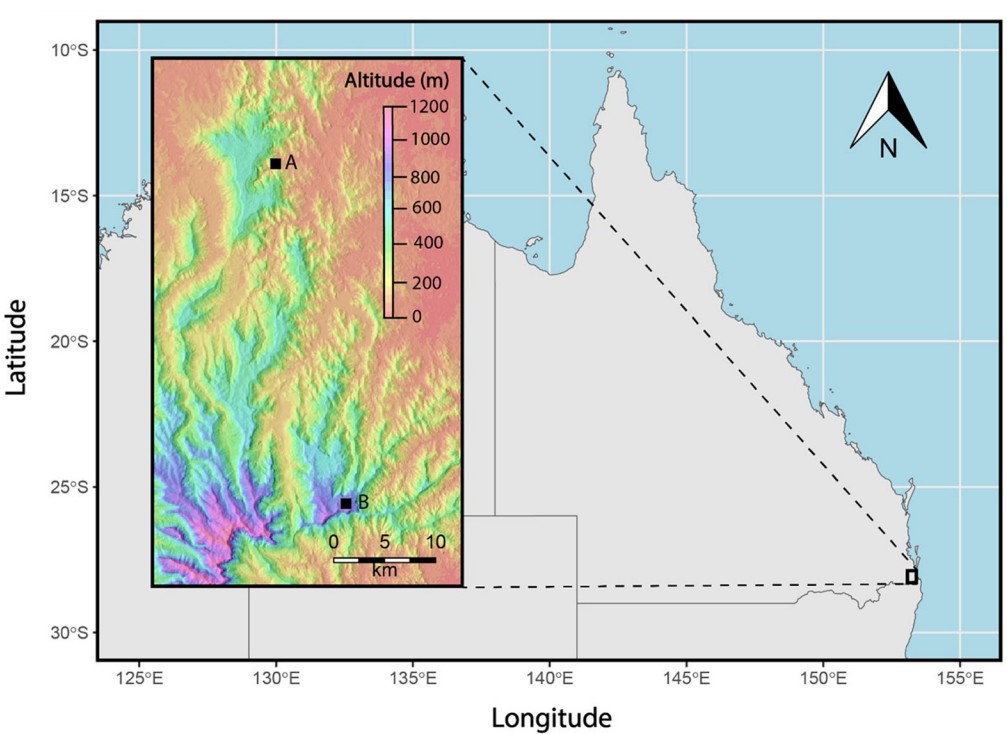

**Fig. 1. Study site locations at (A) Guanaba Creek (Guanaba Indigenous Protected Area) and (B) Boy-Ull Creek (Springbrook National Park), southeast Queensland, Australia.** Sampling was conducted over four periods from October 2022 to March 2023. At each site, independent biological replicates included multiple pools (*n*=2 pools per site) with sensors placed in distinct microhabitats within each pool (*n*=7–13 UV sensors per sampling period; total *n*=39 UV sensors). Site locations were recorded using a handheld GPS unit (coordinates: Guanaba Creek: −27.943296, 153.216153; Boy-Ull Creek: −28.225654, 153.274431). No statistical analyses were applied to the site location data.

course of 5 months (November 2022 to March 2023) on select days at Guanaba Creek (Guanaba Indigenous Protected Area) and Boy-Ull Creek (Springbrook National Park; Table 1). The climate at both sites is temperate, with cool winters and warm summers. Sampling periods were determined by selecting periods of several days where at least one sunny day was forecast at both creeks.

Guanaba Creek is located within the Coomera River catchment in the Gold Coast region. At Guanaba Creek, sampling took place beneath the Tamborine Mountain plateau accessed via Guanaba Indigenous Protected Area (Ngarang-wal Gold Coast Aboriginal Association Inc.). The Guanaba Creek sampling locations were at 'low elevations' (<99 m above sea level) along two adjacent pools on metasediment in Guanaba Creek (coordinates: −27.943296, 153.216153). These sites were in a dry rainforest/wet eucalypt forest dominated by piccabeen palms (*Archontophoenix cunninghamiana*). Pools had little leaf litter and algal mats were sometimes present on the substrate in the creek.

On Springbrook plateau at higher elevations, sampling took place in the Boy-Ull Creek sub-catchment at two sites at higher elevation (775 m above sea level) in adjacent pools (coordinates: −28.225654, 153.274431). Sampled sites at Boy-Ull creek were in rhyolitic soil with heavy leaf litter. Canopy cover was higher compared with Guanaba Creek due to sampling sites being in wet rainforest dominated by black wattle (*Callicoma serratifolia*). Both sampled creeks originate from volcanic mountain

ranges associated with Wollumbin and the Tweed Caldera, sites of high cultural and biological significance. Springbrook Plateau is a hotspot for amphibian biodiversity, hosting a range of rare and threatened species.

**Dosimetry**

Estimation of UV-B radiation (UVBR) dose in amphibian habitats was made using 0.254 mm poly-sulfone UVBR absorbance film dosimeters (Melinex/Mylar 516,100 µm, Archival Survival, Victoria, Australia).

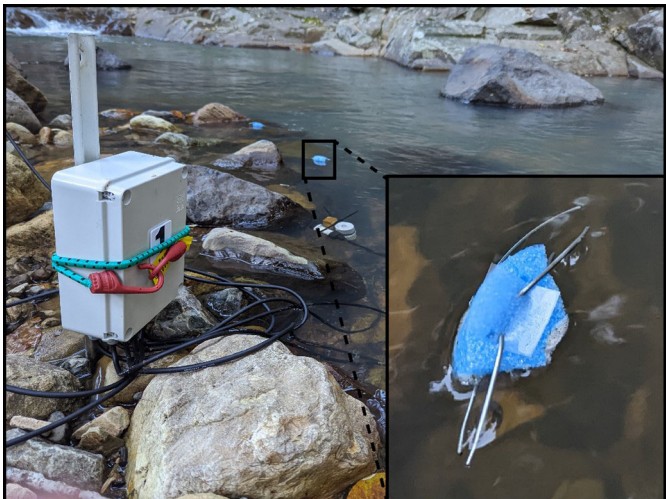

**Fig. 2. Deployment of UVB dosimeters in larval amphibian habitats at Guanaba Creek (Guanaba Indigenous Protected Area) and Boy-Ull Creek (Springbrook National Park).** Dosimeters (*n*=3 per pool, total *n*=12 per site, *n*=24 across both sites) were placed in aquatic mounts floating at the water surface and tethered to the substrate to prevent drift. Dosimeters were deployed at sunrise and retrieved after 1–4 days to estimate cumulative UVB exposure. Each dosimeter was an independent replicate within each site, and absorbance measurements were made in triplicate per dosimeter. Statistical analyses for UVB dose estimates were conducted using two-way ANOVA with site and month as fixed effects (see Fig. 9 for details).

**Table 1. 2022–2023 sampling period at Guanaba Creek (Guanaba Indigenous Protected Area) and Boy-Ull Creek (Springbrook National Park)**

| Sampling dates | Full days sampled | UV sensors | Temperature sensors | Sampling hours |
|---|---|---|---|---|
| 29/10/22–2/11/22 | 1 | 9 | 12 | 216 |
| 23/11/22–27/11/22 | 3 | 7 | 9 | 504 |
| 16/1/23–20/1/23 | 3 | 13 | 13 | 936 |
| 1/3/23–5/3/23 | 4 | 10 | 12 | 960 |
| Total | 11 | 39 | 45 | 2616 |

Only full days of sampling were used in analyses. Sampling hours were calculated as total hours of sampled UVR data from full sampling days only. UVR and temperature data was logged every 15 s and a roughly equal number of sensors were deployed at each site within sampling periods.

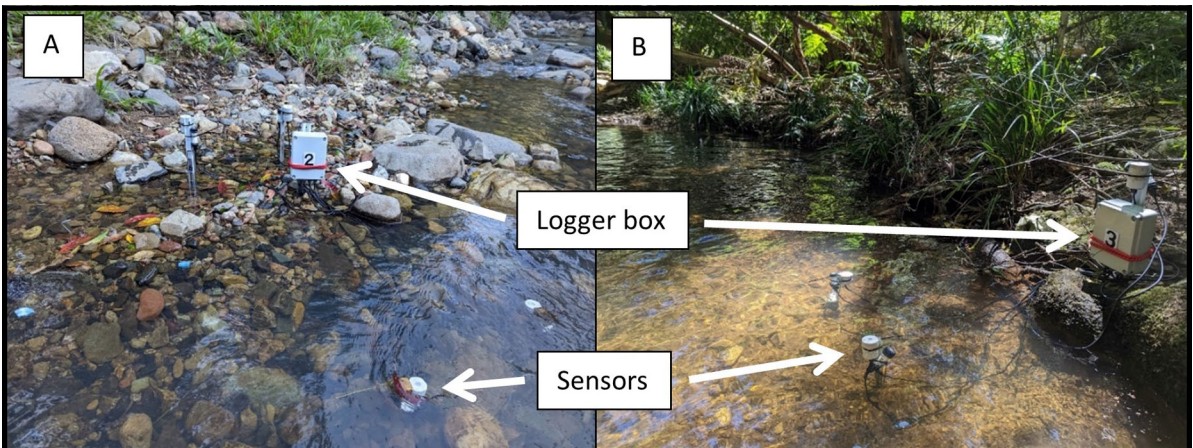

**Fig. 3. Deployment of UV and temperature sensors at study sites in (A) Guanaba Creek (Guanaba Indigenous Protected Area, *n*=7–13 UV sensors, *n*=12 temperature sensors) and (B) Boy-Ull Creek (Springbrook National Park, *n*=7–13 UV sensors, *n*=12 temperature sensors).** Sensors were placed at 1 cm depth beneath the water surface and secured to stakes using zip ties to ensure stable positioning. Each sensor represents an independent biological replicate, capturing UV and temperature variability within each pool. Sensor calibration was conducted prior to deployment (see Materials and Methods). Statistical comparisons of UV and temperature readings were performed using two-way ANOVA (see Fig. 5 and Table 3 for details).

Previous studies have validated polysulfone film as an effective proxy for cumulative UV exposure in ecological contexts (Parisi and Wong, 1994), including in freshwater streams (Frost et al., 2006). The absorbance of polysulfone film increases proportionally with UVBR exposure (Peterson et al., 2002). For field experiments, dosimeters were placed in aquatic mounts floating at the water surface on blue plastic foam supports (Fig. 2), tethered to the substrate to prevent drift. Dosimeters were deployed at sunrise and collected after 1–4 days at sunset to mirror the logger data full day sampling regimes. Within site replication occurred in triplicate.

A cumulative UVB dose-response standard curve was generated using a spectroradiometer (IL1400BL, International Light) in full sunlight. The standard curve was used to generate UVB doses for field measurements of UVBR dosimeter film made using a spectrophotometer (Beckman Coulter DU800, Thermo Fisher Scientific, Waltham, MA, USA) measuring absorbance at 330 nm to estimate UVBR dose received by the film. Daily UVBR dose was averaged from the transformed dose-response values resulting from spectrophotometer analysis of dosimeter films.

### UVR and temperature data loggers

For direct measurement of UVR and temperature in the field, custom data loggers were developed using low cost commercially available components. A full list of items and instructions for logger design, wiring schematics and programming codes are publicly available on UQ eSpace (Hird et al., 2023b). While double monochromator spectroradiometers are considered the standard for UV measurement (Bernhard and Seckmeyer, 1999), their use in field settings is limited by size, cost, and power requirements. Diode-array spectroradiometers, though sensitive to stray light effects (Gao et al., 2017), offer a practical alternative for longer-term *in situ* monitoring. To mitigate these limitations, our study employed rigorous calibration protocols and cross-referenced field data with established reference instruments. These corrections ensure reliable field measurements despite the inherent constraints of diode-array spectroradiometers (Gröbner et al., 2005).

Weatherproof housing was designed which contained a microcontroller, battery supply, data-logger shield, and low power printed circuit board. The houses were fixed to the creek banks. From waterproof glands, 1.5 m 4-core

waterproof wires (*n*=8 per unit) connected to UV and temperature sensors situated in waterproof housing. All UVR sensors were covered by thin 0.1 cm polytetrafluoroethylene (PTFE) filter to allow light penetration through the housing window (Schaar, 2019), and then sealed underneath UV transparent glass. The UVR sensors were positioned vertically by zip ties to stakes 1 cm beneath the surface of the water, with temperature sensors at the same level (Fig. 3). A sensor depth of 1 cm was initially chosen to reduce the likelihood of sedimentation from disturbance of the substrate. Water samples were collected at random times on random days twice for each site and stored in the dark in a cooler at ~4°C on site before being transported to The University of Queensland for analysis within 7 days of collection. Water samples were stored in airtight, opaque containers at ~4°C immediately after collection to prevent photodegradation. Percent UV

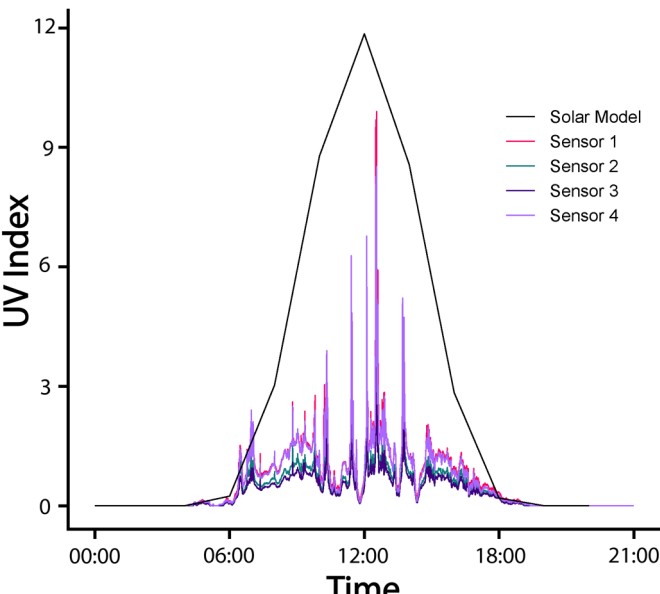

**Fig. 4. Representative sampled UVI readings every 15 s for 24 h on January 20, 2023, at one site (Guanaba Creek; Guanaba Indigenous Protected Area) (*n*=4 UV sensors) compared with the estimated clear sky UVI as predicted by the solar radiative transfer model (black line).** Each sensor represents an independent biological replicate (i.e. distinct locations within the site). Data are presented as raw UVI readings, and no statistical analysis was applied to this dataset.

**Table 2. Percent UVA and UVB transmittance (mean±s.d.) for Guanaba Creek (Guanaba Indigenous Protected Area) and Boy-Ull Creek (Springbrook National Park), across all months**

| Site | 300 nm | 340 nm | 360 nm |
|---|---|---|---|
| Guanaba Creek | 97.1±4.8 | 98.8±2.6 | 99.3±1.9 |
| Boy-Ull Creek | 97.9±2.6 | 99.8±0.5 | 99.5±1.6 |

Biology Open

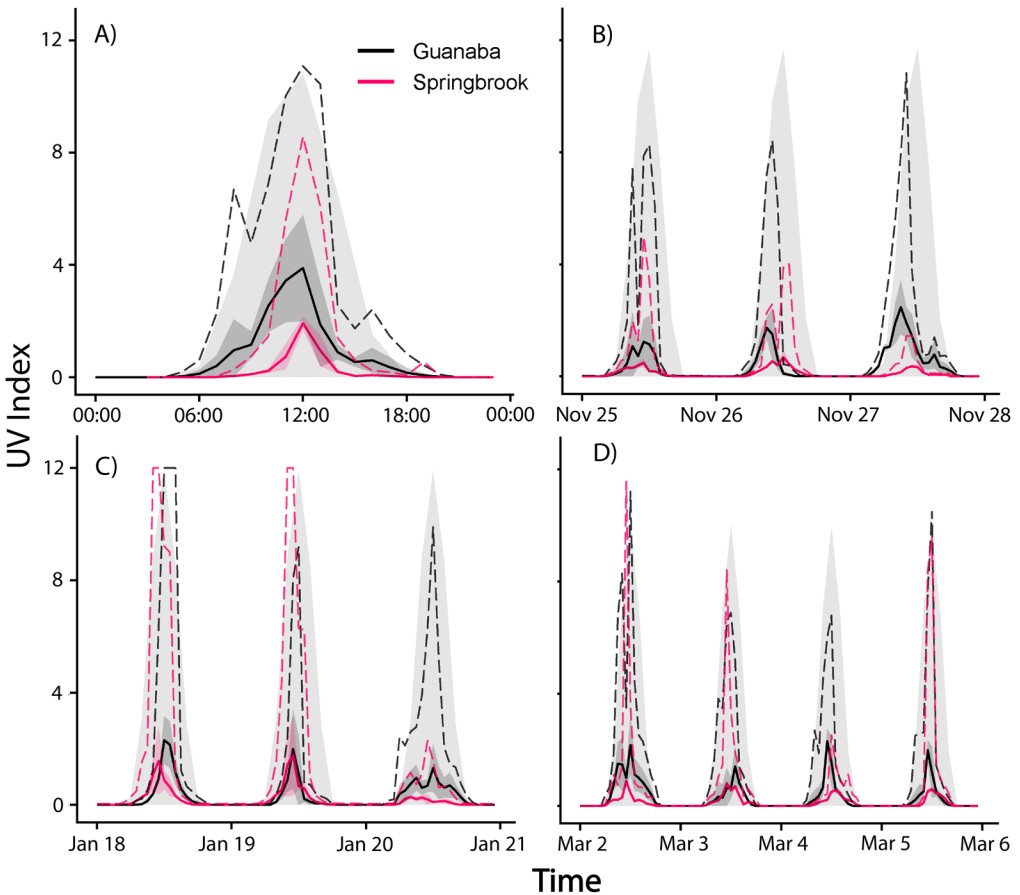

**Fig. 5. Hourly mean UVI (solid lines) ±s.d.** (dark shading) averaged across sensors within Guanaba Creek (Guanaba Indigenous Protected Area; $n$=7–13 UV sensors per period) and Boy-Ull Creek (Springbrook National Park; $n$=7–13 sensors per period) during sampling periods in (A) 31 October 2022, (B) November 2022, (C) January 2023, and (D) March 2023. Dashed lines represent maximum hourly UVI values recorded across sensors. Light grey shading represents the predicted UVI from the solar radiative transfer models. Values were calculated from independent biological replicates (individual sensors placed in distinct microhabitats). UVI values were analysed using a two-way ANOVA with site and month as fixed factors ($F_{3,52}$=4.87, $P$<0.01). *Post hoc* pairwise comparisons were adjusted for multiple comparisons using Tukey's test.

transmittance through 1 cm of water sample was measured for each sample in duplicate at 300, 340, and 360 nm wavelengths in the UV spectrum using UV-transparent cuvettes in a spectrophotometer (Thermo Fisher Scientific). Samples were kept chilled (∼4°C) throughout transport and storage to minimise microbial activity and chemical changes before analysis.

Sensor depth was monitored daily over the sampling periods. The deepest areas of the pools and the water depth at each logger were measured. Percentage of UV irradiance at the benthos was estimated for the maximum measured depth using the Beer–Lambert law, which describes the attenuation of light as it passes through a medium and is commonly used to estimate how much UV radiation penetrates through water at different depths (e.g. Palen et al., 2002). The formula is:

$$I_d = I_0 \cdot e^{-ad},$$

where $I_d$ is the estimated UV irradiance at depth $d$; $I_0$ is the measured UV irradiance at the water surface; $a$ is the attenuation coefficient (in cm$^{-1}$); $d$ is the depth in cm; and $e$ is Euler's number (∼2.718). The attenuation coefficient $a$ was derived from the measured percentage transmittance per 1 cm depth using:

$$a = \frac{\ln(T)}{d},$$

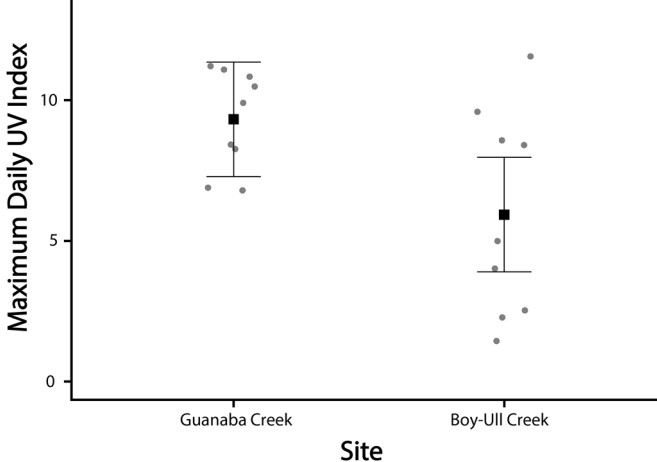

**Fig. 6. Maximum daily UV Index within Guanaba Creek (Guanaba Indigenous Protected Area, *n*=7–13 sensors per period) and Boy-Ull Creek (Springbrook National Park; *n*=7–13 sensors per period) sites averaged across the entire sampling period and all pools sampled within sites.** Circle points are maximum daily UVI readings for each data logger unit and squares are the means. Error bars represent mean±s.d. Statistical comparisons between sites were performed using a one-way ANOVA ($F_{1,16}$=6.24, $P$=0.024).

**Table 3. Average (± s.d.), maximum and minimum daily temperature for a given month at Guanaba Creek (Guanaba Indigenous Protected Area) and Boy-Ull Creek (Springbrook National Park)**

| Month | Site | Temperature (°C) | | |
|---|---|---|---|---|
| | | Average | Maximum | Minimum |
| Guanaba Creek | January | 21.6±0.4 | 25.4 | 17.7 |
| | March | 22.5±0.6 | 26.2 | 19.8 |
| | November | 19.8±0.6 | 23.4 | 17.1 |
| | October | 18.7±0.1 | 20.7 | 16.5 |
| Boy-Ull Creek | January | 17.1±0.4 | 22.2 | 13.3 |
| | March | 17.5±0.3 | 22.9 | 16.6 |
| | November | 15.5±1.2 | 18.0 | 12.1 |
| | October | 16.2±0.4 | 17.9 | 14.4 |

Biology Open

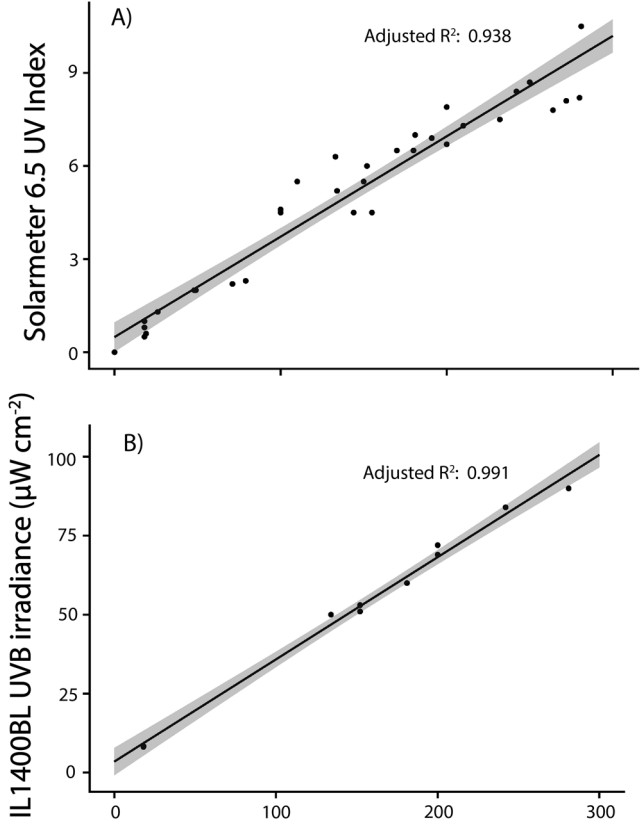

Fig. 7. Correlation between UVB readings made between the Solarmeter UVB radiometer and (A) simultaneous UVI readings, as well as (B) simultaneous UVB readings made on the IL1400BL radiometer. Lines are predictions from the fitted models±95% confidence intervals (grey shading). Points are raw UVR measurements. (A) UVI versus Solarmeter UVB ($F_{1,33}$=514.4, $P$<0.001, $R^2$=0.94) and (B) Solarmeter UVB versus IL1400BL UVB ($F_{1,8}$=943.7, $P$<0.001, $R^2$=0.99). Data points represent independent measurements across multiple time points ($n$=34 for UVI versus Solarmeter UVB; $n$=9 for Solarmeter UVB versus IL1400BL). Regression lines represent the fitted models with 95% confidence intervals (grey shading).

where $T$ is the fractional transmittance (e.g. 97%=0.97) per cm; and $d$=1 cm for the measured transmittance values.

Each adjacent pool at the two study sites spanned approximately 9–15 m in length and 3–4 m in width. With sensors placed at multiple locations spaced approximately 30 cm apart to reflect small-scale spatial heterogeneity in UV exposure caused by canopy cover and local topography. Sensors were positioned near the streambanks and in open-channel areas to capture a range of UVR conditions across the aquatic microenvironment.

UVR sensors for each separate microcontroller unit were calibrated against full spectrum sunlight following Schaar (2019) using a reference UVI meter (Solarmeter 6.5, Solarlight Company Inc., Glenside, USA). Recalibration was done prior to each deployment. UVI and temperature readings were obtained from loggers every ~15 s. The UV-transparent glass above the PTFE filters was wiped daily over the sampling periods with a soft toothbrush to minimise potential interference by biofilm or sediment fouling with UVR detection. Thermal data were collected through the water column at the substrate and 1 cm below the water surface, but no evidence of thermal stratification was evident at the study sites over the course of sampling. All underwater temperature data were therefore assumed to represent the thermal conditions underwater for each of the UVR logger positions. A subset of UVR measurements was made at the water surface each day that loggers were deployed over the sampling periods using handheld spectroradiometers (IL1400BL, International Light Inc., Newburyport,

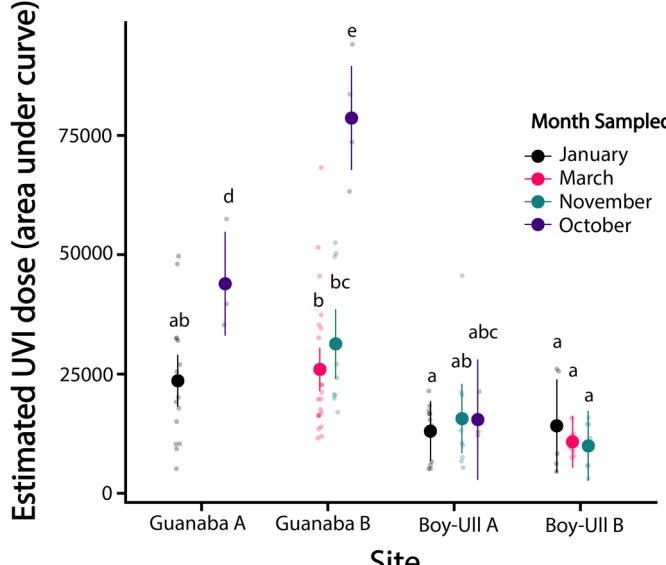

Fig. 8. Daily UVI dose estimated from loggers across all four pools sampled within Guanaba Creek (Guanaba Indigenous Protected Area; $n$=7–13 UV sensors per period) and Boy-Ull Creek (Springbrook National Park; $n$=7–13 UV sensors per period) sites in four different sampling months (October 2022–March 2023). Lowercase letters indicate significantly different groups based on *post hoc* pairwise comparisons following a two-way ANOVA with site and month as interacting factors ($F_{4,99}$=7.12, $P$<0.001). Tukey's test was used to correct for multiple comparisons. Smaller points represent individual daily UVI doses. Smaller points represent individual daily UVI doses, and larger points represent means±s.d.

USA; Solarmeter 6.0, Solarlight Company Inc., Glenside, USA; Solarmeter 6.5, Solarlight Company Inc., Glenside, USA). Solarmeter instruments have sometimes underestimated UVI by 10–20% (de Corrêa et al., 2010), so reported UVI values may be lower than what would have been recorded using research-grade instruments.

### Solar radiative modelling

Because spectroradiometers are not conducive to field deployment in amphibian habitats, there remains a need to understand if radiative transfer models are generally in agreement with *in situ* measurements (Diamond et al., 2005). Radiative transfer models calculate solar irradiance for any time and place while accounting for a range of complex variables that can influence the UVR reaching the ground. Spectral measurements of UVR generally agree with radiative transfer models on cloudless days in the Australian tropics (Bernhard et al., 1997). We modelled UVI at Guanaba Creek and Boy-Ull Creek sites every 2 h over each sampling period using the simple model of the atmospheric radiative transfer of sunshine (SMARTS 2.9.8; Gueymard, 2019). Although UVI is weighted for human erythemal response (McKinlay and Diffey, 1987), we used it primarily as an environmental index rather than a direct measure of biological effects on amphibians. This was modelled under clear-sky conditions using custom input parameters to incorporate effect of location and atmospheric attenuation of UVR. An input file with the appropriate parameters for replication was uploaded to UQ eSpace (doi:10.48610/0f81bde).

### Amphibian surveys

Surveys of adult and larval amphibians were undertaken to ensure deployment of loggers was representative of sites where amphibian larvae, including sensitive species, were distributed. Surveys used the visual encounter search method (Bury and Major, 1997; Thoms et al., 1997). A surveyor (Hird) patrolled shallow water edges of sites for 15 min, searching substrate, shoreline, and macrophytes for species presence (adults, larvae, and eggs). Incidental frog calls were recorded. Sites were surveyed each day and night over the study periods when loggers were deployed.

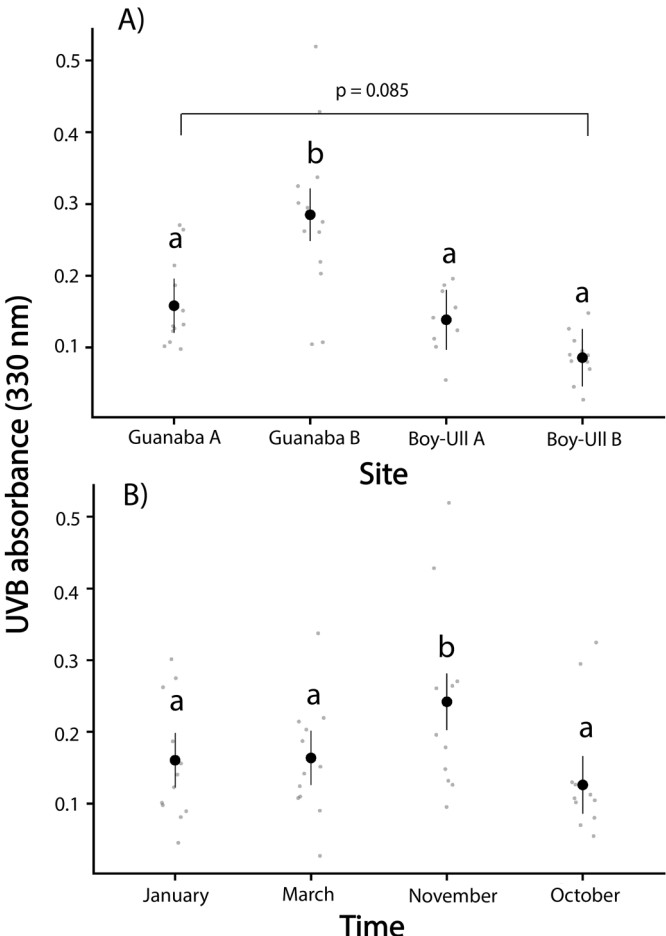

**Fig. 9. Daily UVB dose estimated from dosimeter film across (A) all four pools sampled within Guanaba Creek (Guanaba Indigenous Protected Area; *n*=3 dosimeters per pool, total *n*=12) and Boy-Ull Creek (Springbrook National Park; *n*=3 dosimeters per pool, total *n*=12) sites, and (B) sampled months.** Data represent mean±s.d. Lowercase letters indicate significantly different groups based on *post hoc* pairwise comparisons following a two-way ANOVA with site and month as independent variables ($F_{3,30}$=18.89, *P*<0.001 for site; $F_{3,30}$=6.56, *P*<0.01 for month). Tukey's test was used to correct for multiple comparisons. Smaller points represent individual daily UVB doses measured from dosimeters.

## Statistical analyses

All data were analysed using R (version 4.3.0; R Core Team, 2023). Data were subset to represent only full overlapping sampling days (00:00–23:59) within each site, rather than days on either side of these sampling periods where loggers did not capture data for the entire day. Data were cleaned to remove environmental anomalies unrelated to typical UVR or temperature conditions. Outliers were identified using the interquartile range (IQR) method, where values falling 1.5 times the IQR above the third quartile (Q3) or below the first

quartile (Q1) were flagged as potential outliers. These flagged values were visually inspected, and only those resulting from likely sensor malfunctions (e.g. sudden unrealistic spikes or drops inconsistent with natural UVR variation) were removed. Average, maximum, and minimum hourly UVI values were calculated for each UV sensor. Average, maximum, and minimum daily temperature values were calculated for each underwater temperature sensor. All data used in models passed assumptions of statistical tests used, assessed using the performance package (version 0.10.3; Lüdecke et al., 2021). Alpha was 0.05 for all tests. All *post hoc* analyses compared between groups using the emmeans package (version 1.8.6). Maximum daily UV was modelled with site and month as independent variables in an ANOVA using the car package (version 3.1-2; Fox et al., 2013). This model combined UVR data from both pools within site locations.

Cleaned UVI data were subset into single days and daily UVI dose was calculated as area under curve (AUC) using trapezoidal integration from the pracma package (version 2.4.2) for each UV sensor. Daily AUC data were modelled using site and month as interacting independent variables in a two-way ANOVA using the car package. Average daily UV dose obtained from the dosimetric film was modelled using the same method. Trends and correlations in UVR dose between AUC and dosimetry methods were not compared statistically as not all sites were sampled for each month.

Spectroradiometric UVB and UVI readings taken throughout the study period were modelled using linear regression. Linear regression was also used to correlate UVB readings from the handheld radiometric instruments used. Water transmittance data were compared between sites and wavelengths by fitting an ANOVA model using the car package.

## Acknowledgements
The authors gratefully acknowledge Ashleah, Georgia, and Joshua, for their volunteering support. We acknowledge Felix Meyer whose persistence and expertise was critical in the design of the logger into an application. We acknowledge the assistance of Michael Mayrow in the engineering of the electronics. We thank the custodians of Guanaba Indigenous Protected Area for their enthusiasm in the monitoring of their Country and acknowledge our responsibilities as visitors (Hird et al., 2023c). We particularly thank Justine Dillon for their support. We also acknowledge institutional support from The University of Queensland's technical and research facilities that contributed to this project.

## Competing interests
The authors declare no competing or financial interests.

## Author contributions
Conceptualization: C.H., R.L.C., C.E.F.; Data curation: C.H., R.L.C., C.E.F.; Formal analysis: C.H., R.L.C., C.E.F.; Funding acquisition: R.L.C., C.E.F.; Investigation: C.H., R.L.C., C.E.F.; Methodology: C.H., R.L.C., C.E.F.; Project administration: R.L.C., C.E.F.; Resources: C.E.F.; Software: C.E.F.; Supervision: R.L.C., C.E.F.; Validation: C.H., R.L.C., C.E.F.; Visualization: C.H., R.L.C., C.E.F.; Writing – original draft: C.H.; Writing – review & editing: C.H., R.L.C., C.E.F.

## Funding
This research was financially supported by an Australian Research Council Discovery grant (DP190102152) to C.E.F. All applicable institutional and national guidelines for the care and use of animals were followed. C.H. was a recipient of a Research Training Program (RTP) scholarship and a Ric Nattrass Research Grant from the Queensland Frog Society. Funding sources had no role in study design or the collection, analysis and interpretation of data, in the writing of the report, or in the decision to submit the article for publication. Open Access funding provided by The University of Queensland. Deposited in PMC for immediate release.

## Data and resource availability
The complete datasets and R scripts used for analysing the data are publicly available at UQ eSpace (Hird et al., 2023b).

## Peer review history
The peer review history is available online at https://journals.biologists.com/jcs/article-lookup/doi/10.1242/bio.061991.

## References
**Alados-Arboledas, L., Alados, I., Foyo-Moreno, I., Olmo, F. J. and Alcántara, A.** (2003). The influence of clouds on surface UV erythemal irradiance. *Atmos. Res.* **66**, 273-290. doi:10.1016/S0169-8095(03)00027-9

### Table 4. Amphibian species detected at Guanaba Creek (Guanaba Indigenous Protected Area) and Boy-ull Creek (Springbrook National Park) sampling sites during systematic sampling over the course of the study period

| Site | Species | Life history stage | Detection |
|---|---|---|---|
| Guanaba Creek | *Litoria pearsoniana* | Adult | Visual |
| | *Litoria wilcoxii* | Adult, larval | Visual, auditory |
| | *Rhinella marina* | Adult | Visual |
| Boy-Ull Creek | *Litoria chloris* | Adult, larval | Visual, auditory |
| | *Litoria pearsoniana* | Adult | Visual, auditory |
| | *Mixophyes fasciolatus* | Adult, larval | Visual, auditory |
| | *Mixophyes fleayi* | Adult | Visual, auditory |

**Alroy, J.** (2015). Current extinction rates of reptiles and amphibians. *Proc. Natl. Acad. Sci. USA* **112**, 13003-13008. doi:10.1073/pnas.1508681112

**Alton, L. A. and Franklin, C. E** (2012). Do high temperatures enhance the negative effects of ultraviolet-B radiation in embryonic and larval amphibians? *Biol. Open* **1**, 897-903. doi:10.1242/bio.2012950

**Alton, L. A. and Franklin, C. E.** (2017). Drivers of amphibian declines: effects of ultraviolet radiation and interactions with other environmental factors. *Clim. Change Resp.* **4**, 1-26. doi:10.1186/s40665-017-0034-7

**Angilletta, M. J.** (2009). Thermal acclimation. In M. J. Angilletta (Ed.), *Thermal Adaptation*, pp. 126-156. Oxford University Press.

**Antón, M., Serrano, A., Cancillo, M. L. and García, J. A.** (2009). Experimental and forecasted values of the ultraviolet index in southwestern Spain. *J. Geophys. Res.* **114**, D05211. doi:10.1029/2008JD011304

**Anzalone, C. R., Kats, L. B. and Gordon, M. S.** (2008). Effects of solar UV-B radiation on embryonic development in Hyla cadaverina, Hyla regilla, and Taricha torosa. *Conserv. Biol.* **12**, 646-653. doi:10.1111/j.1523-1739.1998.96478.x

**Aukes, P. J. K., Schiff, S. L., Venkiteswaran, J. J., Elgood, R. J. and Spoelstra, J.** (2021). Size-based characterization of freshwater dissolved organic matter finds similarities within a waterbody type across different Canadian ecozones. *Limnol. Oceanogr.* **6**, 85-95. doi:10.1002/lol2.10180

**Bais, A. F., McKenzie, R. L., Bernhard, G., Aucamp, P. J., Ilyas, M., Madronich, S. and Tourpali, K.** (2015). Ozone depletion and climate change: impacts on UV radiation. *Photochem. Photobiol. Sci.* **14**, 19-52. doi:10.1039/c4pp90032d

**Bais, A. F., Lucas, R. M., Bornman, J. F., Williamson, C. E., Sulzberger, B., Austin, A. T., Wilson, S. R., Andrady, A. L., Bernhard, G., McKenzie, R. L. et al.** (2018). Environmental effects of ozone depletion, UV radiation and interactions with climate change: UNEP Environmental Effects Assessment Panel, update 2017. *Photochem. Photobiol. Sci.* **17**, 127-179. doi:10.1039/c7pp90043k

**Bancroft, B. A., Baker, N. J. and Blaustein, A. R.** (2008a). A meta-analysis of the effects of ultraviolet B radiation and its synergistic interactions with pH, contaminants, and disease on amphibian survival. *Conserv. Biol.* **22**, 987-996. doi:10.1111/j.1523-1739.2008.00966.x

**Bancroft, B. A., Baker, N. J., Searle, C. L., Garcia, T. S. and Blaustein, A. R.** (2008b). Larval amphibians seek warm temperatures and do not avoid harmful UVB radiation. *Behav. Ecol.* **19**, 879-886. doi:10.1093/beheco/arn044

**Barnes, P. W., Flint, S. D. and Caldwell, M. M.** (1987). Photosynthesis damage and protective pigments in plants from a latitudinal Arctic/Alpine gradient exposed to supplemental UV-B radiation in the field. *Arc. Antarct. Alp. Res.* **19**, 21-27. doi:10.1080/00040851.1987.12002573

**Barnes, P. W., Williamson, C. E., Lucas, R. M., Robinson, S. A., Madronich, S., Paul, N. D., Bornman, J. F., Bais, A. F., Sulzberger, B., Wilson, S. R. et al.** (2019). Ozone depletion, ultraviolet radiation, climate change and prospects for a sustainable future. *Nat. Sustain.* **2**, 569-579. doi:10.1038/s41893-019-0314-2

**Belmont, P., Morris, D. P., Pazzaglia, F. J. and Peters, S. C.** (2009). Penetration of ultraviolet radiation in streams of eastern Pennsylvania: topographic controls and the role of suspended particulates. *Aquat. Sci.* **71**, 189-201. doi:10.1007/s00027-009-9120-7

**Berger, L., Speare, R., Daszak, P., Green, D. E., Cunningham, A. A., Goggin, C. L., Slocombe, R., Ragan, M. A., Hyatt, A. D., McDonald, K. R. et al.** (1998). Chytridiomycosis causes amphibian mortality associated with population declines in the rain forests of Australia and Central America. *Proc. Natl. Acad. Sci. USA* **95**, 9031-9036. doi:10.1073/pnas.95.15.9031

**Bernhard, G. and Seckmeyer, G.** (1999). Uncertainty of measurements of spectral solar UV irradiance. *J. Geophys. Res. Atmos.* **104**, 14321-14345. Doi:10.1029/1999JD900180

**Bernhard, G., Mayer, B., Seckmeyer, G. and Moise, A.** (1997). Measurements of spectral solar UV irradiance in tropical-Australia. *J. Geophys. Res. Atmos.* **102**, 8719-8730. doi:10.1029/97JD00072

**Biodiversity Group**. (1999). *Declines and Disappearences of Australian Frogs*. Environment Australia, Department of the Environment and Heritage.

**Blaustein, A. R. and Wake, D. B.** (1990). Declining amphibian populations: a global phenomenom? *Trends Ecol. Evol.* **5**, 4-5. doi:10.1016/0169-5347(90)90129-2

**Blaustein, A. R. and Wake, D. B.** (1995). The puzzle of declining amphibian populations. *Sci. Am.* **272**, 56-61. doi:10.1038/scientificamerican0495-52

**Blaustein, A. R., Romansic, J. M., Kiesecker, J. M. and Hatch, A. C.** (2003). Ultraviolet radiation, toxic chemicals and amphibian population declines. *Divers. Distrib.* **9**, 123-140. doi:10.1046/j.1472-4642.2003.00015.x

**Bradford, D. F.** (1984). Temperature modulation in a high-elevation Amphibian, Rana muscosa. *Copeia* **1984**, 966. doi:10.2307/1445341

**Bradford, D. F.** (1991). Mass mortality and extinction in a high-elevation population of Rana muscosa. *J. Herpetol.* **25**, 174-177. doi:10.2307/1564645

**Brattstrom, B. H.** (1979). Amphibian temperature regulation studies in the field and laboratory. *Am. Zool.* **19**, 345-356. doi:10.1093/icb/19.1.345

**Broomhall, S. D., Osborne, W. S. and Cunningham, R. B.** (2000). Comparative effects of ambient ultraviolet-B radiation on two sympatric species of Australian frogs. *Conserv. Biol.* **14**, 420-427. doi:10.1046/j.1523-1739.2000.98130.x

**Bukaveckas, P. A. and Robbins-Forbes, M.** (2000). Role of dissolved organic carbon in the attenuation of photosynthetically active and ultraviolet radiation in Adirondack lakes. *Freshw. Biol.* **43**, 339-354. doi:10.1046/J.1365-2427.2000.00518.X

**Bury, R. B. and Major, D. J.** (1997). Integrated sampling for amphibian communities in montane habitats. In *Sampling Amphibians in Lentic Habitats: Methods and Approaches for the Pacific Northwest* (ed. D. H. Olson, W. P. Leonard and B. R. Bury), pp. 75-82. Olympia, Washington, USA: Society for Northwestern Vertebrate Biology.

**Caldwell, M. M., Robberecht, R. and Billings, W. D.** (1980). A steep latitudinal gradient of solar ultraviolet-B radiation in the arctic-alpine life zone. *Ecology* **61**, 600-611. doi:10.2307/1937426

**Carey, C.** (1993). Hypothesis Concerning the Causes of the Disappearance of Boreal Toads from the Mountains of Colorado. *Conserv. Biol.* **7**, 355-362. doi:10.1046/j.1523-1739.1993.07020355.x

**Ceccato, E., Cramp, R. L., Seebacher, F. and Franklin, C. E.** (2016). Early exposure to ultraviolet-B radiation decreases immune function later in life. *Conserv. Physiol.* **4**, cow037. doi:10.1093/conphys/cow037

**Cockell, C.** (2000). The ultraviolet history of the terrestrial planets - implications for biological evolution. *Planet. Space. Sci.* **48**, 203-214. doi:10.1016/S0032-0633(99)00087-2

**Cockell, C. S. and Blaustein, A. R.** (2001). *Ecosystems, Evolution, and Ultraviolet Radiation*. New York, USA: Springer Verlag New York Inc.

**Cramp, R. L., Ohmer, M. E. B. and Franklin, C. E.** (2022). UV exposure causes energy trade-offs leading to increased chytrid fungus susceptibility in green tree frog larvae. *Conserv. Physiol.* **10**, coac038. doi:10.1093/CONPHYS/COAC038

**Dahlback, A., Gelsor, N., Stamnes, J. J. and Gjessing, Y.** (2007). UV measurements in the 3000-5000 m altitude region in Tibet. *J. Geophys. Res. Atmos.* **112**, D09308. doi:10.1029/2006JD007700

**De Corrêa, M. P., Godin-Beekmann, S., Haeffelin, M., Brogniez, C., Verschaeve, F., Saiag, P., Pazmiño, A. and Mahé, E.** (2010). Comparison between UV index measurements performed by research-grade and consumer-products instruments. *Photochem. Photobiol. Sci.* **9**, 459-463. doi:10.1039/b9pp00179d

**Di Sarra, A., Cacciani, M., Chamard, P., Cornwall, C., Deluisi, J. J., Di Iorio, T., Disterhoft, P., Fiocco, G., Fuã¡, D. and Monteleone, F.** (2002). Effects of desert dust and ozone on the ultraviolet irradiance at the Mediterranean island of Lampedusa during PAUR II. *J. Geophys. Res. Atmos.* **107**, 2. doi:10.1029/2000JD000139

**Diamond, S. A., Trenham, P. C., Adams, M. J., Hossack, B. R., Knapp, R. A., Stark, S. L., Bradford, D., Corn, P. S., Czarnowski, K., Brooks, P. D. et al.** (2005). Estimated ultraviolet radiation doses in wetlands in six national parks. *Ecosystems* **8**, 462-477. doi:10.1007/s10021-003-0030-6

**Downie, A. T., Wu, N. C., Cramp, R. L. and Franklin, C. E.** (2023). Sublethal consequences of ultraviolet radiation exposure on vertebrates: synthesis through meta-analysis. *Glob. Chang. Biol.* **29**, 6620-6634. doi:10.1111/GCB.16848

**Eppeldauer, G. P.** (2012). Standardization of Broadband UV Measurements for 365 nm LED Sources. *J. Res. Natl. Inst. Stand. Technol.* **117**, 96. doi:10.6028/JRES.117.004

**Fox, J., Friendly, M. and Weisberg, S.** (2013). Hypothesis tests for multivariate linear models using the car package. *R. J.* **5**, 39-52. doi:10.32614/RJ-2013-004

**Frost, P. C., Mack, A., Larson, J. H., Bridgham, S. D. and Lamberti, G. A.** (2006). Environmental controls of UV-B radiation in forested streams of northern Michigan. *Photochem. Photobiol.* **82**, 786. doi:10.1562/2005-07-22-RA-619

**Gao, W., Grant, R. H. and Slusser, J. R.** (2017). *UV Radiation in Global Climate Change: Measurements, Modeling, and Effects on Ecosystems*. Berlin, Heidelberg: Springer.

**Gehrmann, W. H., Horner, J. D., Ferguson, G. W., Chen, T. C. and Holick, M. F.** (2004). A comparison of responses by three broadband radiometers to different ultraviolet-B sources. *Zoo Biol.* **23**, 355-363. doi:10.1002/ZOO.20014

**Gergel, S. E., Turner, M. G. and Kratz, T. K.** (1999). Dissolved organic carbon as an indicator of the scale of watershed influence on lakes and rivers. *Ecol. Appl.* **9**, 1377-1390. doi:10.1890/1051-0761(1999)009[1377:DOCAAI]2.0.CO;2

**Gillespie, G. R., Roberts, J. D., Hunter, D., Hoskin, C. J., Alford, R. A., Heard, G. W., Hines, H., Lemckert, F., Newell, D. and Scheele, B. C.** (2020). Status and priority conservation actions for Australian frog species. *Biol. Conserv.* **247**, 108543. doi:10.1016/j.biocon.2020.108543

**Gröbner, J. and Blumthaler, M., Sugiura, O., Xiang, R., Nakamura, T. and Katagiri, M.** (2008). Stray light correction of diode array spectrometers. *Rev. Sci. Instrum.* **76**, 023102. doi:10.1063/1.1835632

**Gueymard, C. A.** (2019). The SMARTS spectral irradiance model after 25 years: new developments and validation of reference spectra. *Sol. Energy* **187**, 233-253. doi:10.1016/J.SOLENER.2019.05.048

**Häder, D. P., Kumar, H. D., Smith, R. C. and Worrest, R. C** (2007). Effects of solar UV radiation on aquatic ecosystems and interactions with climate change. *Photochem. Photobiol. Sci.* **6**, 267-285. doi:10.1039/b700020k

**Häder, D. P., Williamson, C. E., Wängberg, S., Rautio, M., Rose, K. C., Gao, K., Helbling, E. W., Sinha, R. P. and Worrest, R.** (2015). Effects of UV radiation on aquatic ecosystems and interactions with other environmental factors. *Photochem. Photobiol. Sci.* **14**, 108-126. doi:10.1039/c4pp90035a

**Hays, J. B., Blaustein, A. R., Kiesecker, J. M., Hoffman, P. D., Pandelova, L., Coyle, D. and Richardson, T.** (1996). Developmental responses of amphibians

to solar and artificial UVB sources: a comparative study. *Photochemistry and Photobiology* **64**, 449-456. doi:10.1111/j.1751-1097.1996.tb03090.x

**Hird, C., Franklin, C. E. and Cramp, R. L.** (2022). Temperature causes species-specific responses to UV-induced DNA damage in amphibian larvae. *Biol. Lett.* **18**, 20220358. doi:10.1098/RSBL.2022.0358/

**Hird, C., Cramp, R. L. and Franklin, C. E.** (2023a). Thermal compensation reduces DNA damage from UV radiation. *J. Therm. Biol.* **117**, 103711. doi:10.1016/j.jtherbio.2023.103711

**Hird, C., Cramp, R. L. and Franklin, C. E.** (2023b). The ultraviolet microenvironment in freshwater ecosystems: implications for amphibians. The University of Queensland. Data Collection. doi:10.48610/0f81bde

**Hird, C., David-Chavez, D. M., Spang Gion, S. and Van Uitregt, V.** (2023c). Moving beyond ontological (worldview) supremacy: indigenous insights and a recovery guide for settler-colonial scientists. *J. Exp. Biol.* **226**, jeb245302. doi:10.1242/jeb.245302

**Hof, C., Araújo, M. B., Jetz, W. and Rahbek, C.** (2011). Additive threats from pathogens, climate and land-use change for global amphibian diversity. *Nature* **480**, 516-519. doi:10.1038/nature10650

**Hossack, B. R., Diamond, S. A. and Corn, P. S.** (2006). Distribution of boreal toad populations in relation to estimated UV-B dose in Glacier National Park, Montana, USA. *Can. J. Zool.* **84**, 98-107. doi:10.1139/z05-184

**Houlahan, J. E., Findlay, S. C., Schmidt, B. R., Meyer, A. H. and Kuzmin, S. L.** (2000). Quantitative evidence for global amphibian population declines. *Nature* **404**, 752-755. doi:10.1038/35008052

**Huey, R. B. and Kingsolver, J. G.** (1989). Evolution of thermal sensitivity of ectotherm performance. *Trends Ecol. Evol.* **4**, 131-135. doi:10.1016/0169-5347(89)90211-5

**Humphreys, W. F.** (1978). The thermal biology of Geolycosa godeffroyi and other burrow inhabiting Lycosidae (Araneae) in Australia. *Oecologia* **31**, 319-347. doi:10.1007/BF00346251/METRICS

**Jimenez, I. M., Kühl, M., Larkum, A. W. D. and Ralph, P. J.** (2008). Heat budget and thermal microenvironment of shallow-water corals: do massive corals get warmer than branching corals? *Limnol. Oceanogr.* **53**, 1548-1561. doi:10.4319/LO.2008.53.4.1548

**Kern, P., Cramp, R. L. and Franklin, C. E.** (2014). Temperature and UV-B-insensitive performance in tadpoles of the ornate burrowing frog: an ephemeral pond specialist. *J. Exp. Biol.* **217**, 1246-1252. doi:10.1242/jeb.097006

**Kiesecker, J. M. and Blaustein, A. R.** (1995). Synergism between UV-B radiation and a pathogen magnifies amphibian embryo mortality in nature. *Proc. Natl. Acad. Sci. USA* **92**, 11049-11052. doi:10.1073/pnas.92.24.11049

**Kiesecker, J. M., Blaustein, A. R. and Belden, L. K**. (2001). Complex causes of amphibian population declines. *Nature* **410**, 681-684. doi:10.1038/35070552

**Koepke, P., Reuder, J. and Schwander, H.** (2002). Solar UV radiation and its variability due to the atmospheric components. *Recent Res. Dev. Photochem. Photobiol.* **6**, 11-34.

**Körner, C.** (2003). Global change at high elevation. In C. Körner (Ed.) *Alpine Plant Life: Functional Plant Ecology of High Mountain Ecosystem*, pp. 291-298. Berlin: Springer-Verlag.

**Larason, T. C. and Cromer, C. L.** (2001). Sources of error in UV radiation measurements. *J. Res. Natl. Inst. Stand. Technol.* **106**, 649-656. doi:10.6028/JRES.106.030

**Laurion, I., Ventura, M., Catalan, J., Psenner, R. and Sommaruga, R.** (2000). Attenuation of ultraviolet radiation in mountain lakes: factors controlling the among- and within-lake variability. *Limnol. Oceanogr.* **45**, 1274-1288. doi:10.4319/LO.2000.45.6.1274

**Leech, D. M. and Williamson, C. E.** (2001). In situ exposure to ultraviolet radiation alters the depth distribution of Daphnia. *Limnol. Oceanogr.* **46**, 416-420. doi:10.4319/lo.2001.46.2.0416

**Lemus-Deschamps, L. and Makin, J. K.** (2012). Fifty years of changes in UV Index and implications for skin cancer in Australia. *Int. J. Biometeorol.* **56**, 727-735. doi:10.1007/s00484-011-0474-x

**Lemus-Deschamps, L., Rikus, L. and Gies, P.** (1999). The operational Australian ultraviolet index forecast 1997. *Meteorol. Appl.* **6**, 241-251. doi:10.1017/S1350482799001188

**Licht, L. E.** (2003). Shedding light on ultraviolet radiation and amphibian embryos. *Bioscience* **53**, 551-561. doi:10.1641/0006-3568(2003)053[0551:sloura]2.0.co;2

**Lüdecke, D., Ben-Shachar, M. S., Patil, I., Waggoner, P. and Makowski, D.** (2021). Performance: an R package for assessment, comparison and testing of statistical models. *J. Open Source Softw.* **6**, 3139. doi:10.21105/JOSS.03139

**Lundsgaard, N. U., Cramp, R. L., Franklin, C. E. and Martin, L.** (2020). Effects of ultraviolet-B radiation on physiology, immune function and survival is dependent on temperature: implications for amphibian declines. *Conserv. Physiol.* **8**, coaa002. doi:10.1093/conphys/coaa002

**Lundsgaard, N. U., Cramp, R. L. and Franklin, C. E.** (2021). Ultraviolet-B irradiance and cumulative dose combine to determine performance and survival. *J. Photochem. Photobiol. B* **222**, 112276. doi:10.1016/j.jphotobiol.2021.112276

**Lundsgaard, N. U., Cramp, R. L. and Franklin, C. E.** (2022). Early exposure to UV radiation causes telomere shortening and poorer condition later in life. *J. Exp. Biol.* **225**, jeb243924. doi:10.1242/JEB.243924

**Lundsgaard, N. U., Hird, C., Doody, K. A., Franklin, C. E. and Cramp, R. L.** (2023). Carryover effects from environmental change in early life: an overlooked driver of the amphibian extinction crisis? *Glob. Chang. Biol.* **29**, 3857-3868. doi:10.1111/GCB.16726

**Madronich, S., McKenzie, R. L., Caldwell, M. M. and Bjorn, L. O.** (1995). Changes in ultraviolet radiation reaching the earth's surface. *Ambio* **24**, 143-152. doi:10.2307/4314320

**Markager, S. and Vincent, W. F.** (2000). Spectral light attenuation and the absorption of UV and blue light in natural waters. *Limnol. Oceanogr.* **45**, 642-650. doi:10.4319/LO.2000.45.3.0642

**McCallum, M. L.** (2007). Amphibian decline or extinction? Current declines dwarf background extinction rate. *BioOne* **41**, 483-491. doi:10.1670/0022-1511(2007)41

**McDonald, K. and Alford, R. A.** (1999). A review of declining frogs in northern Queensland. In *Declines and Disappearances of Australian Frogs* (ed. A. Campbell), pp. 14-22. Environment Australia, Department of the Environment and Heritage.

**McKinlay, A. F. and Diffey, B. L.** (1987). A reference action spectrum for ultraviolet induced erythema in human skin. *CIE J.* **6**, 17-22.

**McKenzie, R. L., Aucamp, P. J., Bais, A. F., Björn, L. O., Ilyas, M. and Madronich, S.**, (2020). Ozone depletion and climate change: impacts on UV radiation. *Photochem. Photobiol. Sci.* **10**, 182-198. doi:10.1039/c0pp90034f

**Montzka, S. A., Dutton, G. S., Yu, P., Ray, E., Portmann, R. W., Daniel, J. S., Kuijpers, L., Hall, B. D., Mondeel, D., Siso, C. et al.** (2018). An unexpected and persistent increase in global emissions of ozone-depleting CFC-11. *Nature* **557**, 413-417. doi:10.1038/s41586-018-0106-2

**Morison, S. A., Cramp, R. L., Alton, L. A. and Franklin, C. E.** (2020). Cooler temperatures slow the repair of DNA damage in tadpoles exposed to ultraviolet radiation: implications for amphibian declines at high altitude. *Glob. Chang. Biol.* **26**, 1225-1234. doi:10.1111/gcb.14837

**Morris, D. P., Zagarese, H., Williamson, C. E., Balseiro, E. G., Hargreaves, B. R., Modenutti, B., Moeller, R. and Queimalinos, C.** (1995). The attenuation of solar UV radiation in lakes and the role of dissolved organic carbon. *Limnol. Oceanogr.* **40**, 1381-1391. doi:10.4319/lo.1995.40.8.1381

**Palen, W. J., Schineider, D. E., Adams, M. J., Pearl, C. A., Bury, R. B. and Diamond, S. A.** (2002). Optical characteristics of natural waters protect amphibians from UV-B in the U.S. Pacific northwest. *Ecology* **83**, 2951-2957. doi:10.1890/0012-9658(2002)083[2951:OCONWP]2.0.CO;2

**Parisi, A. V. and Wong, J. C.** (1994). A dosimetric technique for the measurement of ultraviolet radiation exposure to plants. *Photochem. Photobiol.* **60**: 470-474. doi:10.1111/j.1751-1097.1994.tb05136.x

**Peng, S., Liao, H., Zhou, T. and Peng, S.** (2017). Effects of UVB radiation on freshwater biota: a meta-analysis. *Glob. Ecol. Biogeogr.* **26**, 500-510. doi:10.1111/GEB.12552

**Peterson, G. S., Johnson, L. B., Axler, R. P. and Diamond, S. A.** (2002). Assessment of the risk of solar ultraviolet radiation to amphibians. II. In situ characterization of exposure in amphibian habitats. *Environ. Sci. Technol.* **36**, 2859-2865. doi:10.1021/ES011196L

**Pfeifer, M. T., Koepke, P. and Reuder, J.** (2006). Effects of altitude and aerosol on UV radiation. *J. Geophys. Res. Atmos.* **111**, 1-11. doi:10.1029/2005JD006444

**Pincebourde, S., Murdock, C. C., Vickers, M. and Sears, M. W.** (2016). Fine-scale microclimatic variation can shape the responses of organisms to global change in both natural and urban environments. *Integr. Comp. Biol.* **56**, 45-61. doi:10.1093/ICB/ICW016

**R Core Team**. (2023). R: A language and environment for statistical computing.

**Rader, R. B. and Belish, T. A.** (1997). Effects of ambient and enhanced UV-B radiation on periphyton in a mountain stream. *J. Freshw. Ecol.* **12**, 615-628. doi:10.1080/02705060.1997.9663576

**Richards, S. J., McDonald, K. R. and Alford, R. A.** (1993). Declines in populations of Australia's endemic tropical rainforest frogs. *Pac. Conserv. Biol.* **1**, 66-77. doi:10.1071/PC930066

**Schaar, R.** (2019). *Designing the VEML6075 Into an Application*. Vishay Semiconductors. Vishay, PE, USA.

**Schindler, D. W., Bayley, S. E., Curtis, P. J. et al.** (1992). Natural and man-caused factors affecting the abundance and cycling of dissolved organic substances in precambrian shield lakes. In *Dissolved Organic Matter in Lacustrine Ecosystems: Energy Source and System Regulator* (eds. E. M. Perdue and E. T. Gjessing), pp. 1-21. Netherlands: Springer.

**Scully, N. M. and Lean, D. R. S.** (1994). The attenuation of ultraviolet radiation in temperate lakes. *Limnol. Oceanogr.* **39**, 651-661.

**Sommaruga, R. and Psenner, R.** (1997). Ultraviolet radiation in a high mountain lake of the Austrian Alps: air and underwater measurements. *Photochem. Photobiol.* **65**, 957-963. doi:10.1111/J.1751-1097.1997.TB07954.X

**Stelzner, J. K. and Hausfater, G.** (1986). Posture, microclimate, and thermoregulation in yellow baboons. *Primates* **27**, 449-463. doi:10.1007/BF02381890/METRICS

**Stuart, S. N., Chanson, J. S., Cox, N. A., Young, B. E., Rodrigues, A. S., Fischman, D. L. and Waller, R. W.** (2004). Status and trends of amphibian declines and extinctions worldwide. *Science (1979)* **306**, 1783-1786. doi:10.1126/science.1103538

**Thoms, C., Corkran, C. C. and Olson, D. H.** (1997). Basic amphibian survey for inventory and monitoring in lentic habitats. In *Sampling Amphibians in Lentic Habitats: Methods and Approaches for the Pacific Northwest* (ed. H. D. Olsen, P. W. B. Leonard and R. Bury), pp. 35-46. Olympia, Washington, USA: Society for Northwestern Vertebrate Biology.

**Ultsch, G. R., Bradford, D. F. and Freda, J.** (1999). Physiology coping with the environment. In *Tadpoles: The Biology of Anuran Larvae* (ed. R.W. McDiarmid and R. Altig), pp. 202-210. Chicago: University of Chicago Press.

**Van Uitregt, V. O., Wilson, R. S. and Franklin, C. E.** (2007). Cooler temperatures increase sensitivity to ultraviolet B radiation in embryos and larvae of the frog Limnodynastes peronii. *Glob. Chang. Biol.* **13**, 1114-1121. doi:10.1111/j.1365-2486.2007.01353.x

**Vanicek, K., Frei, T., Lytinska, Z. and Schmalwieser, A.** (2000). *UV-index for the Public, Report of the COST-713 Action*. Brussels, Belgium: European Commission.

**Wang, Q. W., Hidema, J. and Hikosaka, K.** (2014). Is UV-induced DNA damage greater at higher elevation? *Am. J. Bot.* **101**, 796-802. doi:10.3732/ajb.1400010

**Williamson, C. E., Stemberger, R. S., Morris, D. P., Frost, T. M. and Paulsen, S. G.** (1996). Ultraviolet radiation in North American lakes: attenuation estimates from DOC measurements and implications for plankton communities. *Limnol. Oceanogr.* **41**, 1024-1034. doi:10.4319/lo.1996.41.5.1024

**Williamson, C. E., Zepp, R. G., Lucas, R. M., Madronich, S., Austin, A. T., Ballare, C. L., Norval, M., Sulzberger, B., Bais, A. F., McKenzie, R. L. et al.** (2014). Solar ultraviolet radiation in a changing climate. *Nat. Clim. Chang.* **4**, 434-441. doi:10.1038/nclimate2225

**Wollmuth, L. P. and Crawshaw, L. I.** (1988). The Effect of Development and Season on Temperature Selection in Bullfrog Tadpoles. *Physiol. Zool.* **61**, 461-469. doi:10.1086/physzool.61.5.30161268

**World Meteorological Organization**. (2022). Scientific Assessment of Ozone Depletion: 2022. GAW Report 278. Report No. 278, 509 pp. Geneva, Switzerland. ISBN: 978-9914-733-97-6. Available at: https://library.wmo.int/records/item/58360-scientific-assessment-of-ozone-depletion-2022

**Xenopoulos, M. A. and Schindler, D. W.** (2001). Physical factors determining ultraviolet radiation flux into ecosystems. In *Ecosystems, Evolution, and Ultraviolet Radiation* (ed. C. S. Cockell and A. R. Blaustein), pp. 36-62. New York: Springer.

**Young, B. E., Lips, K. R., Reaser, J. K., Ibáñez, R., Salas, A. W., Cedeño, J. R., Coloma, L. A., Ron, S., La Marca, E., Meyer, J. R. et al.** (2001). Population declines and priorities for amphibian conservation in Latin America. *Conserv. Biol.* **15**, 1213-1223. doi:10.1111/J.1523-1739.2001.00218.X

**Zhu, M., Chen, W. and Chen, L.** (2021). Underwater spectrophotometer for *in-situ* seawater COD monitoring. *Proc. SPIE Ocean Sens. Monitor. XIII*, **11752**, 72-80. doi:10.1117/12.2588022

