## [Peer Review File · Biology Open]

Sunlight surveillance: A simplified approach for the monitoring of harmful UV radiation in freshwater ecosystems

Coen Hird; Rebecca L. Cramp; Craig E. Franklin

DOI: 10.1242/bio.061991

Editor: Lewis Halsey

Review timeline

Original submission: 16 February 2025

Editorial decision: 25 February 2025

Resubmission: 17 March 2025

Accepted: 18 March 2025

Original submission

First decision letter

MS ID#: bio.061940

MS Title: Sunlight surveillance: A simplified approach for the monitoring of harmful UV radiation in freshwater

Authors: Coen Hird; Rebecca L Cramp; Craig E Franklin

Dear Dr Hird,

I am writing to let you know that I have now reached a decision on the above manuscript. I am afraid that, after careful consideration, I feel that it cannot currently be accepted for publication in Biology Open.

The reviewer reports are shown at the bottom of this email or can be accessed, together with a copy of this decision letter, by going to:

As you will see, one of the reviewers in particular raises a number of substantial criticisms that prevent me from accepting your paper for publication. Overall, the conclusions are not adequately supported by the data, the logger cannot reasonably be argued to be gold standard as the authors claim, and the loggers being positioned at a very shallow depth under the water is unlikely to be optimal. Moreover, I agree with one of the Reviewers that week-old water will have changed in composition.

I realise that this is disappointing news, and we understand the frustration that you must feel. However, I am sure that you appreciate that the conclusions of your research must be seen by the wider community to be fully supported by the data. On this occasion, I have decided that this is not the case.

I do hope you find the reviewer comments helpful in allowing you to revise the manuscript for successful submission to Biology Open or elsewhere.

Biology Open is committed to high-quality production and a fair, constructive and efficient editorial process. To help us achieve these aims, we would be very grateful if you could take a few minutes to complete our author survey.

Comments from the Reviewers:

Reviewer 1: The study by Hird et al. aimed to quantify UV radiation (UVR) in two freshwater streams in Australia, with the premise that UV radiation is harmful to aquatic life, especially amphibian eggs and larvae (tadpoles). The authors imply that amphibian declines could be due to unknown effects of altered UVR caused by anthropogenic environmental changes, but that this is unknown because the equipment to quantify UVR has been mostly unavailable or excessively expensive. The authors therefore designed what they refer to as 'novel underwater UVR loggers' (line 17). They claim that "fine scale UVR data logging was proven to be a gold standard for monitoring the UVR microenvironment in freshwater systems" (lines 24-25).

I realise that this is disappointing but I found the manuscript well written, and I enjoyed reviewing it, but I do think the authors are overstating their findings and the relevance for aquatic life based on the study they have done. First, I do not understand how they can claim that their equipment/study proved to be the 'gold standard', given that they themselves identify several potential issues with their equipment and approach (lines 175-181, 203-205, 286-287).

Also, the 'underwater' UVR loggers were positioned only 1 cm below the water surface (line 188), which is unlikely to be representative of the primary habitat occupied by the amphibian eggs and larvae. The authors do "acknowledge that UV attenuation is wavelength-dependent and that even in clear water, deeper regions may receive substantially lower UVB levels than surface layers. The assumption that surface measurements approximate conditions at the benthos should therefore be interpreted cautiously, particularly at the most energetic UVB wavelengths, where reported variability in transmittance (Table 2) suggests the potential for significant attenuation at depth" (lines 267-272). Thus, I find it odd why the authors did not place the loggers deeper, which there is no mention of in the manuscript, and I unfortunately think it is a rather large limitation to the study and the conclusions that can be drawn from it. Additionally, in Table 4, the authors present which amphibian species and life stages were observed at the different sites, but there is no information about what water depth these were observed at. Presumably the authors have this information, and it ought to be included in the table so readers can further judge the relevance of measuring UVR at 1 cm depth for amphibians in streams. Did any animals actually reside at 1 cm depth?

Moreover, the authors verify UV transmittance in water from the two streams using spectrophotometry (with stream water in a 1 cm cuvette), but up to seven days after water collection (line 229). As dissolved organic matter, for example, is degraded by light and microbial activity, it seems unlikely that week-old samples are representative of water properties in the field where UVR was measured. I of course appreciate the logistical issues with field work in remote areas, but the authors ought to at least address this rather large time gap between water collection and analysis, and whether any efforts were made to prevent changes in water optical properties.

Specific comments:

Line 36: Wavelengths for UCV are incorrect (typo) here.

Line 43: 'ASL' is not defined yet.

Lines 45-46: The authors state here that "UVR can penetrate significantly into aquatic systems." It would be good to give a depth range or example(s).

Line 47: 'DOM' has not been defined.

Line 61: One or more references ought to be included for the statement that "... [UVR] has been implicated in the global amphibian extinction crisis."

Line 79: The sentence structure here implies that both eggs and tadpoles are 'laid', but it is of course only the eggs that are laid.

Line 133: It is stated here that the study was done over a period of five months, but with no mention of which months. This information should be included.

Lines 142-143: The numbers here are presumably coordinates, but it is unclear.

Line 152: There is an 'and' too many here.

Line 155: 'UVBR' has not been defined (although it is intuitive what it is based on other definitions).

Line 228: I think the word 'esky' is only used in Australia.

Lines 235-236: The authors state here that "Data were cleaned to remove outlier values" but there is no mention of how and based on what criteria this was done. This information needs to be provided.

Reviewer 2: The authors present an innovative approach for monitoring of in situ UV radiation (UVR) levels in freshwater habitats, comparing these measurements with conventional approaches. The study is timely, as amphibian populations are facing global declines from multiple stressors, and understanding realistic UVR exposure is an important but understudied component of freshwater ecology. The manuscript is generally well written, employs appropriate statistical analyses, and provides a compelling demonstration that high-temporal-resolution measurements can detect short bursts of potentially harmful UV irradiance that coarser methods miss. Overall, the paper is of good quality, the methodology is mostly described in sufficient detail, and the results are likely to be of broad interest to ecologists, amphibian biologists, and conservation managers. Below are more specific comments.

1) While the paper provides helpful details on how the authors built and calibrated their cost-effective UV sensors, I feel like there could be much more technical detail included for reproducibility. I think that others that are interested in building their own sensors would have a fairly difficult time based on the depth of details provided here. Perhaps this could be included as a supplement with diagrams.

2) Although the authors clearly state that they cleaned the logger windows daily to avoid fouling, more thoughts on how these loggers might perform over longer deployments (or at remote sites with less frequent upkeep) would be helpful. Some mention of strategies for extended field use (e.g., automated wipers? anti-fouling coatings?) would strengthen the discussion as directions for further development.

3) Given that short-distance changes in canopy cover often result in fine-scale differences in UV exposure, it would be helpful to see more explicit discussion of microhabitat heterogeneity. For instance, do shallow margins or open channels in the same pool receive considerably more or less UV? Such detail (or even speculative discussion) would add ecological context to larval habitat choice and potential refugia.

4) In general I had a very difficult time conceiving the spatial scale over which the sensors were deployed. Additional maps with sampling locations could help here.

5) In the intro the authors mention the roles of cloud cover, canopy shading, and solar angle on the variability of UV. Are there any weather data for the sampling times available that could help interpret day-to-day fluctuations in the logger data?

6) The discussion would benefit from a few more sentences on potential direct physiological mechanisms (e.g., DNA damage, immunosuppression) that might be triggered by these high short-duration UV doses.

7) The paper states that cloud cover, extreme weather, and vegetation shifts could alter UV in freshwater habitats. Briefly elaborating on whether future climate projections anticipate more frequent intense sunshine episodes (or changes in canopy structure) would expand the importance of these data for forecasting amphibian population responses.

Minor:

43: ASL is defined later as meaning above sea level but I would avoid abbreviating this. The paper already has a lot of abbreviations that slow reading.

47: DOM is not defined.

66: not clear why Australia is specified here.

88: Not clear what "This" in "This was" is referring to.

Reviewer's Responses to Questions

Experimental quality

Does each figure have the proper controls?

If 'No', please indicate reasons in Comments for Author box below.

Reviewer #1:

- Yes

Reviewer #2:

- Yes

Were the data analyzed using appropriate statistical tests?

If 'No', please indicate reasons in Comments for Author box below.

Reviewer #1:

- Yes

Reviewer #2:

- Yes

Reproducibility

Were experiments performed using adequate number of biological replicates?

If 'No', please indicate reasons in Comments for Author box below.

Reviewer #1:

- Yes

Reviewer #2:

- Yes

Does the methods section provide sufficient detail to permit reproducibility?

If 'No', please indicate reasons in Comments for Author box below.

Reviewer #1:

- Yes

Reviewer #2:

- Yes

Completeness

Are the manuscript's conclusions supported by the data?

If 'No', please indicate reasons in Comments for Author box below.

Reviewer #1:

- No

Reviewer #2:

- Yes

Scholarship

Do the authors cite and discuss the merits of data that would argue for and against their conclusion?

If 'No', please indicate reasons in Comments for Author box below.

Reviewer #1:

- Yes

Reviewer #2:

- Yes

Does the manuscript title & abstract accurately reflect the contents of the manuscript, without hyperbole?

If 'No', please indicate reasons in Comments for Author box below.

Reviewer #1:

- Yes

Reviewer #2:

- Yes
-

Resubmission

Author response to reviewers' comments

BiO.061991

Sunlight surveillance: A simplified approach for the monitoring of harmful UV radiation in freshwater

Reviewer 1

The study by Hird et al. aimed to quantify UV radiation (UVR) in two freshwater streams in Australia, with the premise that UV radiation is harmful to aquatic life, especially amphibian eggs and larvae (tadpoles). The authors imply that amphibian declines could be due to unknown effects of altered UVR caused by anthropogenic environmental changes, but that this is unknown because the equipment to quantify UVR has been mostly unavailable or excessively expensive. The authors therefore designed what they refer to as 'novel underwater UVR loggers' (line 17). They claim that "fine scale UVR data logging was proven to be a gold standard for monitoring the UVR microenvironment in freshwater systems" (lines 24-25).

I found the manuscript well written, and I enjoyed reviewing it, but I do think the authors are overstating their findings and the relevance for aquatic life based on the study they have done. First, I do not understand how they can claim that their equipment/study proved to be the 'gold standard', given that they themselves identify several potential issues with their equipment and approach (lines 175-181, 203-205, 286-287).

Also, the 'underwater' UVR loggers were positioned only 1 cm below the water surface (line 188), which is unlikely to be representative of the primary habitat occupied by the amphibian eggs and larvae. The authors do "acknowledge that UV attenuation is wavelength-dependent and that even in clear water, deeper regions may receive substantially lower UVB levels than surface layers. The assumption that surface measurements approximate conditions at the benthos should therefore be interpreted cautiously, particularly at the most energetic UVB wavelengths, where reported variability in transmittance (Table 2) suggests the potential for significant attenuation at depth" (lines 267-272). Thus, I find it odd why the authors did not place the loggers deeper, which there is no mention of in the manuscript, and I unfortunately think it is a rather large limitation to the study and the conclusions that can be drawn from it. Additionally, in Table 4, the authors present which amphibian species and life stages were observed at the different sites, but there is no information about what water depth these were observed at. Presumably the authors have this information, and it ought to be included in the table so readers can further judge the relevance of measuring UVR at 1 cm depth for amphibians in streams. Did any animals actually reside at 1 cm depth?

Moreover, the authors verify UV transmittance in water from the two streams using spectrophotometry (with stream water in a 1 cm cuvette), but up to seven days after water collection (line 229). As dissolved organic matter, for example, is degraded by light and microbial activity, it seems unlikely that week-old samples are representative of water properties in the field where UVR was measured. I of course appreciate the logistical issues with field work in remote areas, but the authors ought to at least address this rather large time gap between water collection and analysis, and whether any efforts were made to prevent changes in water optical properties.

- We agree with the reviewer that there is no basis for the claim that our loggers provide a gold standard for UVR measurement and have removed this from the article. We have softened and reworded the article throughout to accurately reflect the study's contributions without overstating its impact, e.g.: "a cost-effective approach for monitoring the UVR microenvironment in freshwater systems. While no single method can be considered universally optimal, our results highlighted the advantages of continuous in situ UV monitoring over traditional methods, particularly for capturing short-duration UV fluctuations relevant to aquatic organisms." Lines 25-29
- We appreciate the reviewer's concern regarding the placement of UVR sensors at 1 cm depth and have revised the manuscript to clarify our reasoning and address potential limitations. We now explicitly justify this choice in the Methods by stating that it reduced the chances of sedimentation on the sensors due to disturbance of the substrate and providing Beer-Lambert Law calculations to estimate UV attenuation at deeper depths. These calculations demonstrate that at 20 cm depth, UVA remains at 78-96% and UVB at 55-65% of surface irradiance, confirming that much of the water column experiences similar UV exposure to surface conditions. Given that amphibian oviposition sites were typically <20 cm deep, our 1 cm sensor placement was appropriate for capturing ecologically relevant UVR conditions. Additionally, we acknowledge in the Discussion that while UVB attenuation is more substantial at the maximum depths of our pools at 40 cm (~30-43% of surface irradiance), UVA likely remains high throughout the water column. Our measurements, therefore, provide a conservative estimate of worst-case UV exposure risks for aquatic organisms. To further address the reviewer's concerns, we now recognise the value of future studies deploying multi-depth sensors to refine depth-dependent UV exposure profiles, particularly for benthic environments. These changes ensure that our findings are well contextualised, and the limitations of surface UVR measurements are appropriately acknowledged. Lines 196, 207-223, 296-305, 375-392
- All larval amphibians recorded in visual detection surveys were found at depths < 20 cm, including depths < 5 cm, and tadpoles were observed surfacing. This is included now on lines 339-341
- We acknowledge the reviewer's concern regarding the potential for changes in water optical properties between collection and spectrophotometric analysis, particularly due to degradation of dissolved organic matter (DOM) by microbial activity and light exposure. To address this, we have revised the Methods section to explicitly state the measures taken to preserve sample integrity. We now clarify on lines 196-201 that:
 - o Water samples were stored in airtight, opaque containers immediately after collection to prevent photodegradation.
 - o Samples were kept chilled (~4 °C) throughout transport and storage to minimise microbial activity and chemical changes before analysis.
 - o Duplicate measurements were taken per sample to ensure consistency in transmittance values.
- In the Discussion, we now acknowledge that some degree of change in optical properties over time is possible but given the low DOM content of these clear-water systems, we expect only minimal alteration in UV transmittance. We also note that similar post-collection storage methods have been used in previous UVR studies in aquatic environments (e.g., (Laurion *et al.*, 2000; Palen *et al.*, 2002; Belmont *et al.*, 2009; Aukes *et al.*, 2021). Lines 387-392

Specific comments:

- Line 36: Wavelengths for UCV are incorrect (typo) here.
 - o This has been corrected to 100-280 nm
- Line 43: 'ASL' is not defined yet.
 - o Abbreviations for ASL were removed following comments from Reviewer 2
- Lines 45-46: The authors state here that "UVR can penetrate significantly into aquatic systems." It would be good to give a depth range or example(s).
 - o "In clear waters, UVB has been recorded penetrating to depths of 10-20 meters, while more turbid or high dissolved organic matter (DOM) environments, attenuation can occur within the first few centimeters to meters (Scully & Lean,

- 1994; Häder et al., 2015). For instance, in some freshwater lakes, UVA can reach depths of 5-30 meters, whereas UVB penetration is typically limited to 1-5 meters depending on DOM concentrations (Morris *et al.*, 1995).” Added to lines 48-52
- Line 47: 'DOM' has not been defined.
 - o This has been corrected
 - Line 61: One or more references ought to be included for the statement that "... [UVR] has been implicated in the global amphibian extinction crisis."
 - o Added references to this line (Blaustein et al., 1994; Blaustein et al., 2003; Häder et al., 2015).
 - Line 79: The sentence structure here implies that both eggs and tadpoles are 'laid', but it is of course only the eggs that are laid.
 - o Revised to read: “Increased UVR is hypothesised to influence amphibian populations by negatively affecting eggs and tadpoles. Amphibian eggs are often laid in shallow water with high UV exposure, and tadpoles are typically diurnal, active during spring and summer when UVR levels are highest. Additionally, both life stages have limited ability to avoid UVR, making them particularly vulnerable”
 - Line 133: It is stated here that the study was done over a period of five months, but with no mention of which months. This information should be included.
 - o Added “(November 2022 -March 2023)”
 - Lines 142-143: The numbers here are presumably coordinates, but it is unclear.
 - o “coordinates:” added at lines 149 and 154
 - Line 152: There is an 'and' too many here.
 - o Revised to “Both sampled creeks originate from volcanic mountain ranges associated with Wollumbin and the Tweed Caldera, sites of high cultural and biological significance. Springbrook Plateau is a hotspot for amphibian biodiversity, hosting a range of rare and threatened species.” On lines 157-160
 - Line 155: 'UVBR' has not been defined (although it is intuitive what it is based on other definitions).
 - o Definition provided: “UV-B radiation”
 - Line 228: I think the word 'esky' is only used in Australia.
 - o Changed to “cooler”
 - Lines 235-236: The authors state here that “Data were cleaned to remove outlier values” but there is no mention of how and based on what criteria this was done. This information needs to be provided.
 - o More detail provided on lines 263-268: “Data were cleaned to remove environmental anomalies unrelated to typical UVR or temperature conditions. Outliers were identified using the interquartile range (IQR) method, where values falling 1.5 times the IQR above the third quartile (Q3) or below the first quartile (Q1) were flagged as potential outliers. These flagged values were visually inspected, and only those resulting from likely sensor malfunctions (e.g., sudden unrealistic spikes or drops inconsistent with natural UVR variation) were removed.”

Reviewer 2

The authors present an innovative approach for monitoring of in situ UV radiation (UVR) levels in freshwater habitats, comparing these measurements with conventional approaches. The study is timely, as amphibian populations are facing global declines from multiple stressors, and understanding realistic UVR exposure is an important but understudied component of freshwater ecology. The manuscript is generally well written, employs appropriate statistical analyses, and provides a compelling demonstration that high-temporal-resolution measurements can detect short bursts of potentially harmful UV irradiance that coarser methods miss.

Overall, the paper is of good quality, the methodology is mostly described in sufficient detail, and the results are likely to be of broad interest to ecologists, amphibian biologists, and conservation managers. Below are more specific comments.

Specific comments:

- While the paper provides helpful details on how the authors built and calibrated their cost-effective UV sensors, I feel like there could be much more technical detail included for reproducibility. I think that others that are interested in building their own sensors would have a fairly difficult time based on the depth of details provided here. Perhaps this could be included as a supplement with diagrams.

- See lines 179-181 which state that “a full list of items and instructions for logger design, wiring schematics and programming codes are publicly available on UQ espace”
- Although the authors clearly state that they cleaned the logger windows daily to avoid fouling, more thoughts on how these loggers might perform over longer deployments (or at remote sites with less frequent upkeep) would be helpful. Some mention of strategies for extended field use (e.g., automated wipers? anti-fouling coatings?) would strengthen the discussion as directions for further development.
 - We have included further discussion around extended field use at lines 468-470 which discusses automated wipers and anti-fouling coatings.
- Given that short-distance changes in canopy cover often result in fine-scale differences in UV exposure, it would be helpful to see more explicit discussion of microhabitat heterogeneity. For instance, do shallow margins or open channels in the same pool receive considerably more or less UV? Such detail (or even speculative discussion) would add ecological context to larval habitat choice and potential refugia.
 - We have expanded discussion to discuss microhabitat heterogeneity, including suggesting future research on larval habitat selection and measuring UV attenuation throughout the water column at lines 362-367 (“Fine-scale spatial variation in UV exposure is an important but often overlooked factor in aquatic ecosystems. While our study focused on UV measurements at a single depth (1 cm), we recognise that UVR exposure likely varies significantly within each pool due to differences in canopy cover and water depth. For example, shallow margins, open channels, and sunlit patches within the same pool may receive considerably higher UV doses than shaded microhabitats beneath overhanging vegetation or within deeper sections of the water column.”) and lines 375-378 (“Future studies should integrate spatially explicit UV mapping with direct observations of amphibian microhabitat use to determine whether larvae preferentially occupy low-UV refugia or are inadvertently exposed to harmful UV doses.”)
- In general I had a very difficult time conceiving the spatial scale over which the sensors were deployed. Additional maps with sampling locations could help here.
 - The stream locations were provided in figure 1 and described in the methods as sampling adjacent pools. We have added further details of the spatial information regarding the sampled pools on lines 219-223: “Each adjacent pool at the two study sites spanned approximately 9-15 m in length and 3-4 meters in width. With sensors placed at multiple locations spaced approximately 30 cm apart to reflect small-scale spatial heterogeneity in UV exposure caused by canopy cover and local topography. Sensors were positioned near the streambanks and in open-channel areas to capture a range of UVR conditions across the aquatic microenvironment.”
- In the intro the authors mention the roles of cloud cover, canopy shading, and solar angle on the variability of UV. Are there any weather data for the sampling times available that could help interpret day-to-day fluctuations in the logger data?
 - We have added discussion of this point to lines 404-409: “Weather data were not directly recorded at our study sites; however, we obtained local weather data, including daily global solar exposure and cloud cover, from the Canungra Defence weather station. While this dataset provided some insight into broader atmospheric conditions, it did not resolve fine-scale differences between our two creek sites. This limitation underscores the importance of site-specific cloud cover monitoring in future studies to better account for localised variability in UVR exposure.”
- The discussion would benefit from a few more sentences on potential direct physiological mechanisms (e.g., DNA damage, immunosuppression) that might be triggered by these high short-duration UV doses.
 - We have included discussion of this at lines 372-374: “It is well established that high-intensity UVR exposure can cause a range of physiological impacts in amphibian larvae, including DNA damage, immunosuppression, and oxidative stress (Blaustein et al., 2003; Morison et al., 2019).”
- The paper states that cloud cover, extreme weather, and vegetation shifts could alter UV in freshwater habitats. Briefly elaborating on whether future climate projections anticipate more frequent intense sunshine episodes (or changes in canopy structure) would expand the importance of these data for forecasting amphibian population responses

- We have included discussion on this point at **lines 440-451**: “Future climate projections suggest that changes in cloud cover, extreme weather events, and vegetation dynamics will significantly alter UV exposure in freshwater habitats. Climate models indicate that many regions, including parts of Australia, are expected to experience more frequent and prolonged periods of intense sunlight due to shifts in atmospheric circulation and reduced cloud cover (Williamson et al. 2014; McKenzie et al. 2020). Additionally, vegetation dieback due to drought and increased wildfire frequency may lead to greater canopy openness, further amplifying UV exposure in aquatic ecosystems (Barnes et al. 2019). These environmental changes could exacerbate UV stress on amphibian populations, particularly for species reliant on shaded microhabitats or those already facing other climate-related stressors such as temperature fluctuations and disease susceptibility. Understanding the fine-scale UVR dynamics in freshwater systems, as explored in this study, is therefore critical for predicting how amphibian populations may respond to future environmental change.”
 - Minor:
 - 43: ASL is defined later as meaning above sea level but I would avoid abbreviating this. The paper already has a lot of abbreviations that slow reading
 - **Removed all ASL abbreviations.**
 - 47: DOM is not defined.
 - **Added definition**
 - 66: not clear why Australia is specified here.
 - **Removed reference to Australia**
 - 88: Not clear what "This" in "This was" is referring to.
 - **Referring to the negative effects of UVR on amphibian health in previous sentence - this was qualified to read “These negative effects” on **line 95****
-

Second decision letter

MS ID#: bio.061991

MS Title: Sunlight surveillance: A simplified approach for the monitoring of harmful UV radiation in freshwater ecosystems

Authors: Coen Hird; Rebecca L Cramp; Craig E Franklin

Dear Dr Hird,

I am happy to tell you that your manuscript has been accepted for publication in Biology Open, pending our standard publication integrity checks.